

# The 2018 northern European hydrological drought and its drivers in a historical perspective

Sigrid J. Bakke[1], Monica Ionita[2], and Lena M. Tallaksen[1]

[1]Department of Geosciences, University of Oslo, Oslo, Norway
[2]Alfred Wegener Institute Helmholtz Centre for Polar and Marine Research, Bremerhaven, Germany

**Correspondence:** Sigrid J. Bakke (s.j.bakke@geo.uio.no)

**Abstract.** In 2018, large parts of northern Europe were affected by an extreme drought. A better understanding of the characteristics and the large-scale atmospheric circulation driving such events is of high importance to enhance drought forecasting and mitigation. This paper examines the historical extremeness of the May–August 2018 meteorological situation and the accompanying meteorological and hydrological (streamflow and groundwater) drought. Further, it investigates the relationship between

the large-scale atmospheric circulation and summer streamflow in the Nordic region. In May and July 2018, record-breaking temperatures were observed in large parts of northern Europe associated with blocking systems centred over Fennoscandia and sea surface temperature anomalies of more than $3°$ C in the Baltic Sea (May, July) and the Barents Sea (July). Extreme meteorological drought, as indicated by the three-month standard precipitation index (SPI3) and precipitation-evapotranspiration index (SPEI3), was observed in May, and covered large parts of northern Europe by July. Streamflow drought in the Nordic region

started to develop in June, and in July 68 % of the stations had record-low or near-record-low streamflow. Extreme streamflow conditions persisted in the southeastern part of the region throughout 2018. Many groundwater wells had record-low or near-record-low levels in July and August. However, extremeness in groundwater levels and (to a lesser degree) streamflow show a diverse spatial pattern. This points to the role of local terrestrial processes in controlling the hydrological response to meteorological conditions, including aquifer properties. Composite analysis of low summer streamflow and 500 mb geopotential

height anomalies revealed a distinction between summer streamflow variability in western/northern Norway and the rest of the region. Low summer streamflow in western/northern Norway is related to high-pressure systems centred over the Norwegian Sea. In the rest of the Nordic region, low summer streamflow is associated with a high-pressure system over the North Sea and a low-pressure system over Greenland and Russia at similar latitudes, resembling the pattern of 2018. This study provides new insight into different hydro-meteorological aspects of the 2018 northern European drought, as well as identification of

large-scale atmospheric circulation patterns associated with summer streamflow drought in the Nordic region.

## 1   Introduction

From May and throughout the summer of 2018, the northern and parts of central Europe experienced drought and record-breaking and persistent high temperatures leading to a variety of severe impacts (Table 1). Drought is a complex phenomenon characterised by below average natural water availability, and unlike most other natural hazards, it is a "creeping phenomenon"



with a wide range of economic, societal, and environmental impacts gradually accumulating over time and space (Stahl et al., 2016; Mishra and Singh, 2010; Tallaksen and Van Lanen, 2004).

In 2018, wild fires destroyed vast areas in northern and central Europe. Sweden was especially impacted, with record breaking 24,310 ha (835 % of average) of burnt area (Table 1,a). The drought also led to significant drop in EU cereal production,

whereas beef production grew more than expected due to increased slaughter following fodder shortage (Table 1,b). In Scandinavia and Germany, wheat and barley yields were described as catastrophically low (Table 1,c–f). Ecosystems in northern Europe are less adapted to extremely dry conditions than other European regions, and direct negative impacts on terrestrial ecosystems productivity were both significantly stronger and and more widespread in 2018 compared to the more southerly centred extreme drought in 2003 (Buras et al., 2020). Already in June, the water volumes in Nordic reservoirs for hydropower

dropped well below normal, and together with high fuel prices it caused the July–August power rates to be the highest in 20 years (Table 1,g,h). Record low river levels disrupted main inland waterways, forcing transportation ships to reduce their loads by up to 85 % (Table 1,i,j). Low water levels in the river Elbe exposed World War 2 munitions (Table 1,k) and so-called hunger stones with centuries old low water level marks along with dire warnings (Table 1,l). Extremely low streamflow and high river temperatures led to fishing bans in major salmon fishing rivers in Norway (Table 1,m). Low groundwater tables led Swedish

municipalities to ban residents from using water from the municipal network for anything other than drinking (Table 1,n). The high costs and wide range of impacts associated with the 2018 drought emphasise the need to improve the understanding of such extreme, high impact events affecting large regions in Europe. The latter requires transnational data and international collaboration for an in-depth analysis.

To understand how the severity and timing of impacts vary among and within drought affected areas, it is important to

distinguish between different stages of drought development. Typically, three types of drought are distinguished, reflecting the propagation of drought through the hydrological cycle; meteorological, soil moisture and hydrological (streamflow and groundwater) drought (Tallaksen and Van Lanen, 2004). Meteorological drought refers to a precipitation deficit often combined with abnormal high (potential) evapotranspiration. If a meteorological drought is sustained, it typically causes soil moisture drought, which mainly concern soil moisture deficits in the root zone impacting water uptake by vegetation (Van Loon, 2015).

When soil moisture depletes, a positive feedback loop might occur due to reduced latent heat flux, making more energy available for sensible heat flux, which in turn increases the near-surface temperature (Seneviratne et al., 2010). Soil moisture drought can further reduce groundwater recharge and water sources that feed streams and rivers. This may, depending on the catchment characteristics and initial hydrological conditions, lead to groundwater and streamflow drought (Tallaksen and Van Lanen, 2004). Several studies have demonstrated how meteorological and hydrological droughts develop differently in

space and time (e.g. Barker et al., 2016; Kumar et al., 2016; Haslinger et al., 2014; Vidal et al., 2010; Tallaksen et al., 2009; Peters et al., 2003; Changnon, 1987). The delay between a meteorological and a hydrological drought may amount to several months, with groundwater typically being the last to react and the last to recover (Hisdal and Tallaksen, 2000). The concept *drought*, when used without specification, refers broadly to the multifaceted phenomenon that includes all three types of drought, along with their different development and nature.



Many large-scale studies on drought focus on the meteorological aspect, such as anomalies in precipitation or climatic water balance (i.e. precipitation minus potential evapotranspiration), as this is based on data often easily at hand (e.g. Ionita et al., 2017; Stagge et al., 2017; Vicente-Serrano et al., 2014; Bordi et al., 2009). As opposed to meteorological data, transboundary near-real-time observations of hydrological variables is generally lacking, making timely observation-based, large-scale soil

moisture, streamflow or groundwater drought assessments challenging (Liu et al., 2018; Laaha et al., 2016; Hannah et al., 2011). Long-term observational soil moisture data is sparse except for satellite based estimates that only cover a few centimetres depth (Hirschi et al., 2014; Kerr, 2007), which is too shallow to include the root zones of main vegetation types (e.g. Yang et al., 2016; Schenk and Jackson, 2002). Data of updated streamflow and groundwater level usually needs to be collected in a country-by-country based manner, which is time-consuming as well as challenging due to differences in agency structure, data

quality requirements, availability of physiographic properties and information of human influence. Despite these challenges, research on large-scale droughts cannot rely solely on meteorological data (Van Lanen et al., 2016). Drought assessments using hydrological data are needed to investigate the drought footprint on water resources, which is of high importance for hydropower, navigation, water use sectors and freshwater ecosystems among others (Laaha et al., 2016; Stahl et al., 2016).

Among the natural drivers of drought are persistent high-pressure systems leading to prolonged periods of low precipitation

and/or high evapotranspiration (Tallaksen and Van Lanen, 2004). To increase our knowledge of how drought characteristics might change in the future, we therefore need a better understanding of the relation between the different types of drought and large-scale atmospheric and oceanographic drivers. Stationary Rossby waves have been found to play an important role in the development of summer patterns of monthly surface temperature and precipitation variability across northern Eurasia, and appear to have led to the extreme heat wave and drought in 2003 and 2010 (Schubert et al., 2014, 2011). Kingston et al.

(2015) found that the most widespread and long-duration meteorological droughts in Europe fall into two categories; northern European droughts with onset associated with an Atlantic meridional dipole atmospheric circulation anomaly similar to the North Atlantic Oscillation (NAO), and droughts elsewhere in Europe associated with anomalies related to a northeastward expansion of the Azores high resembling eastern Atlantic/western Russia (EA/WR) atmospheric circulation patterns. Fleig et al. (2011) investigated the relation between various circulation types and streamflow drought in Denmark and Great Britain,

and found that hydrological droughts were most frequently linked to circulation types representing a high-pressure system over the region affected by drought, which promote hydrological drought development by advection of warm dry air. In addition to stationary high and low-pressure systems, sea surface temperatures associated with large-scale climate modes of variability are also found to be important drivers for dryness and wetness variability over Europe (Ionita et al., 2015, 2012). In a study of streamflow drought in Great Britain, Kingston et al. (2013) found statistically significant SST and atmospheric anomalies

linked to drought onset. The study emphasises the shortcomings in the ability of circulation indices (e.g. NAO) to capture fully the atmospheric variation preceding drought onsets, and highlights the value of composite analysis in developing understanding of ocean-atmosphere-drought connections.

The 2018 event was unique in its northern location of the high-pressure system as compared to other major European drought events in the last decades (Ionita et al., 2017; Stahl, 2001). The affected Nordic region (Norway, Denmark, Sweden and Finland)

exhibits a high heterogeneity in terrestrial and hydroclimatological characteristics. Despite its rather limited size, the region





spans several latitudes and has a pronounced west-east gradient in climate and topography, ranging from high mountains in the west to low-lying regions in the south and east. Prevailing westerly winds run northeastwards from the Atlantic, bringing abundant rainfall in the western part. Orographic effect causes large local variability in precipitation in the mountainous areas. Denmark, southern Sweden, and western coast of Norway have a maritime climate, in contrast to the more continental climate

in eastern Norway, Sweden and Finland. The landscape is largely affected by last glaciations, with typical landforms such as U-shaped valleys, fjords, and lakes and large spatial heterogeneity in glacial deposits. Land cover varies from vast areas of bare rock and shallow deposits in the west and north, to undulating inland areas characterised by numerous lakes, forests and wetlands, and to areas in the south with thick soils and large aquifers (e.g. Sømme, 1960). Combined with the important effect of seasonal snow on hydrology, varying with latitude and altitude, excluding the very south, the result is a high diversity in

hydroclimatological conditions.

In depth analyses of historical drought events, what triggers them and how they manifest themselves in the hydrological cycle, enables us to increase our understanding of this complex phenomenon, which is vital to enhance drought forecasting and mitigation. Motivated by these considerations, this paper focuses on characterizing the 2018 drought in northern Europe in detail, including exploring atmosphere-drought connections. The aim is twofold; 1) to investigate the extremeness of the 2018

situation and the accompanying meteorological and hydrological drought in northern Europe, and 2) to identify large scale atmospheric circulations associated with below normal summer streamflow in the Nordic region.

The paper is organised as follows: The data and methods are described in Sect. 2 and 3, respectively. In Sect. 4 the results shown and described for the 2018 meteorological situation (Sect. 4.1), meteorological drought (Sect. 4.2) and hydrological drought (Sect. 4.3), as well as the relation between summer streamflow and large-scale atmospheric circulation (Sect. 4.4). A

detailed discussion is provided in Sect. 5, followed by the conclusion in Sect. 6.

## 2 Data

### 2.1 Meteorological data

Meteorological data used in this study comprise the 500 mb geopotential height (HGT500), the zonal and meridional wind, sea surface temperature (SST), temperature and precipitation. Monthly data of HGT500, and zonal and meridional wind, used to

describe the atmospheric circulation, were extracted from the NCEP-NCAR 40-year reanalysis project (Kalnay et al., 1996). These datasets are available from 1948 to near-present, and have a global coverage on a $2.5° \times 2.5°$ longitude/latitude grid. SST data was extracted from the National Centers for Environmental Information (NOAA) high resolution Optimum Interpolation Sea Surface Temperature version 2 (OISSTv2; Reynolds et al., 2007). OISSTv2 consists of monthly SST from September 1981 to near-present on a global scale with a spatial resolution of $0.25° \times 0.25°$ longitude/latitude.

Europe-wide (35.625–71.875° N and -10.875–41.625° E) daily total precipitation and daily maximum, minimum and mean temperature on a 0.25 °regular latitude/longitude grid (used for the analysis described in Sect. 3.1 and 3.2), and daily total precipitation on a 0.1 °regular latitude/longitude grid (used for the analysis described in Sect. 3.3), were derived from the





E-OBS dataset version 19.0e (Cornes et al., 2018). The E-OBS datasets are based on the European Climate Assessment and Dataset station information (ECA&D), and consist of daily data from 01.01.1950 until near-present.

## 2.2 Hydrological data

Hydrological data consists of time series of streamflow and groundwater levels from stations in the Nordic region. Streamflow measured at a given point reflects the accumulated responses to precipitation over space and time, whereas groundwater levels represent the lagged response in groundwater over an area varying with local conditions. Streamflow data stem from gauges in Norway, Sweden, Denmark and Finland. Quality-controlled daily observational streamflow time series was provided by the Norwegian Water Resources and Energy Directorate (NVE) for Norway, Danish Environment Portal for Denmark, Swedish Meteorological and Hydrological Institute (SMHI) for Sweden, and Finnish Environmental Institute (SYKE) for Finland. All gauges had near-natural catchments, i.e. the streamflow is to a large degree unaffected by human interventions such as reservoirs or water abstractions. Only gauges having less than 10 days with missing values between May–September each year in the 60-year period 01.01.1959–31.12.2018, were chosen.

The resulting dataset consists of time series from 79 gauges, with catchment areas ranging from 6.6 km$^2$ to 10864 km$^2$ (median of 276 km$^2$). Figure 1 shows the locations of the gauges as well as their annual cycles and streamflow regimes. The streamflow regimes were based on the regime classification of Gottschalk et al. (1979) and calculated for the period 1959–2018 (a detailed description of the classification procedure is provided in Appendix A1). The five regimes reflect the typical streamflow variability in time, classified according to whether the streamflow is dominated by winter high flow and summer low flow, mainly due to high evapotranspiration during summer (Atlantic regime), winter low flow and spring high flow, due to snow accumulation and snowmelt (Mountain regime), or various combinations of these two patterns (Baltic, Transition and Inland regime). Three of the stations with a mountain regime (marked with crosses) experience high flow during late summer due to the large presence of glaciers (>30 % of the catchment).

Observational time series of near-natural groundwater levels, i.e. data from stations with limited or no human influence (such as water abstractions), are even less accessible than streamflow data. This includes the necessary metadata with local site information. As a result, the groundwater analysis was limited to data from stations in Norway and Sweden, provided by NVE and the Geological Survey of Sweden (SGU), respectively. The time series were quality controlled at the host institutions, however, a visual inspection was performed to delete potential erroneous outliers. Groundwater level time series were generally shorter than the streamflow time series, and rather than a 60-year period as used for streamflow, a 30-year period (1989–2018) was selected as a balance between the number of stations and the record length.

The majority of the groundwater wells had weekly to monthly temporal resolution in most of the period covered. In Norwegian wells, daily or sub-daily measurements started at the beginning of the 21st century. Half of the Swedish wells had daily or sub-daily measurements from 2016 onwards, whereas the other half had coarser temporal resolution for the whole period. Only groundwater stations with at least one monthly measurement during April–September over the analysis period, were selected. The varying temporal resolution of the original measurements might affect the results. However, we argue that groundwater in many cases have a slow response and thus have valuable information content at the monthly resolution (e.g.



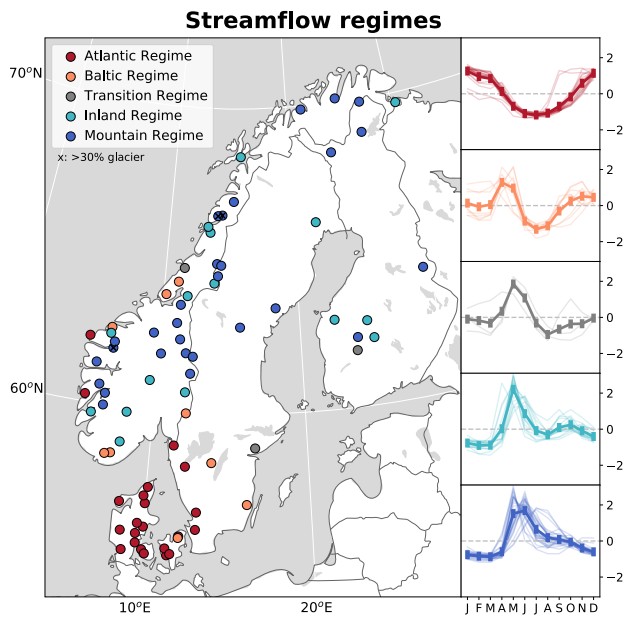

**Figure 1.** Locations and streamflow regimes (based on Gottschalk et al., 1979) of the 79 streamflow stations used in the study. The right panel shows plots of mean monthly standardized (i.e. subtracted the mean and divided by the standard deviation) hydrographs for each regime (indicated by thin lines) together with the regime mean hydrograph (bold line).

Hisdal and Tallaksen, 2000). Sub-daily measurements were aggregated into daily averages, whereas days of missing data were filled by linear interpolation between the two adjacent measurements, following the method used by the National Hydrological Monitoring Programme in the UK (NHMP, 2017).

The final groundwater dataset includes data from 56 wells. Their locations, annual cycles and groundwater regimes are shown in Fig. 2. The groundwater regime classification is based on the classification by Kirkhusmo (1988) using data for the period 1989-2018 (a detailed description of the classification procedure is provided in Appendix A2). Region I is characterised by low groundwater levels in late summer due to warm season evapotranspiration losses. Region III has a minima in late winter prior to the start of the snowmelt period, whereas Region II being a combination of the two, experiences two minima, one in late winter and one in late summer. Some of the wells are classified as a delayed version of a regime due to slow-responding groundwater fluctuations.

## 3   Methods

The variables, indices (including periods used) and spatial coverages used to characterise the 2018 meteorological situation, meteorological drought and hydrological drought are summarised in Table 2. Starting from a large spatial domain including

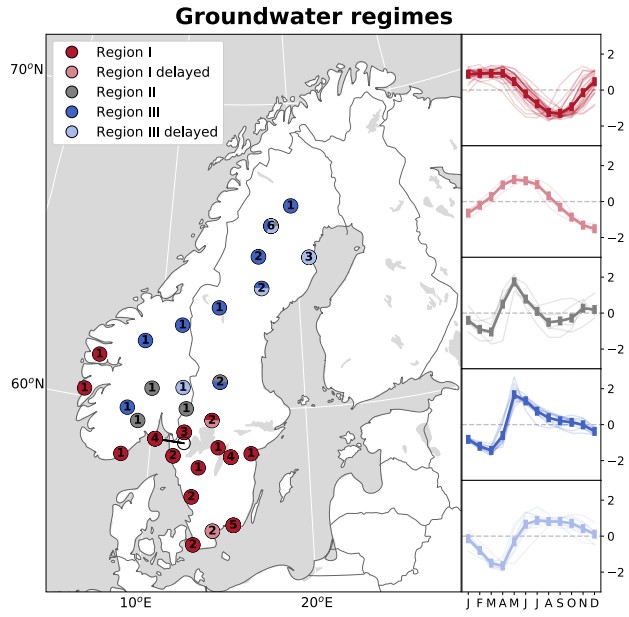

**Figure 2.** Locations and groundwater regimes (based on Kirkhusmo, 1988) of the 56 groundwater wells used in the study. The number on each point represents the number of stations at that location. To ease readability, one site with four wells in southwestern Sweden (red point on the map) is shifted to the left of, and pointing to, its real location. The right panel shows plots of mean monthly standardized (i.e. subtracted the mean and divided by the standard deviation) hydrographs for each regime (indicated by thin lines) together with the regime mean hydrograph (bold line).

Europe and its surrounding regions when describing main climate drivers, the analysis gradually "zooms in" on the Nordic region that shows the most extreme meteorological situation in spring and summer of 2018.

## 3.1 Meteorological Situation

The extremeness of the meteorological situation for each month in May–August 2018 was analysed using the sea surface
5    temperature (SST), geopotential height at 500 mb (HGT500), daily maximum surface temperate (Tx) and precipitation (P). For HGT500 and SST, the 2018 anomalies (in meters and degree Celsius, respectively) relative to a reference period were computed for each month May–August over Europe and the surrounding regions. For HGT500 we used the reference period 1971–2000, whereas for SST we used the reference period from the start of the dataset (in 1981) to 2000. In addition, average May–August HGT500 60-year (1959–2018) time series and corresponding 2018 anomalies (in standard deviations from the
10    60-year mean) were computed for each subdomain of $20° \times 20°$ longitude/latitude throughout the European domain, i.e. the area $35° N$–$80° N$ and $12.5° W$–$42.5° E$ moving one grid cell $(2.5°)$ at a time. This allowed the extremeness in the persistent high-pressure system for the whole May–August period to be estimated.



For temperature and precipitation, monthly mean of Tx and monthly total sums of P were computed for the 60-year period 1959–2018, and monthly ranking maps of 2018 made for the six highest temperature and six lowest precipitation. A ranking map for a specific year (here 2018) is made by first extracting the value of a variable of interest for each year in a chosen period (here 1959–2018), order the sample from the most extreme to the least extreme value, and find the position (rank) of the specific

year. Similar maps were computed for the European 2015 drought by Ionita et al. (2017) using the period 1950–2015. In case of ties between years, 2018 was set as the least extreme of the years with equal values. This was done to avoid exaggerating the extremeness of 2018, such as in some Mediterranean regions where it is not uncommon with months with zero precipitation. A rank of one imply record breaking high temperature (in the case of Tx) or low precipitation (in the case of P) in 2018, a rank of two indicates that 2018 had the second most extreme value in that month, etc. Here, temperature/precipitation with ranks of

1–6 are referred to as extreme. The ranks correspond to specific percentiles of the data, such that a rank of 3 or 6 correspond to the 5th or 10th percentile, respectively, when the period under investigation is 60 years.

### 3.2  Meteorological Drought indices

The meteorological drought of each month May–August 2018 was assessed using the Standardized Precipitation Index (SPI; McKee et al., 1993; Guttman, 1999) and the Standardized Precipitation Evapotranspiration Index (SPEI; Vicente-Serrano et al.,

2010; Beguería et al., 2014). A three-months accumulation period was chosen for both cases (i.e. SPI3 and SPEI3) to reflect the seasonality in northern Europe (WMO, 2012).

SPI is recommended as a meteorological drought index for drought monitoring by the World Meteorological Organization (WMO and GWP, 2016). It is a widely used measure of precipitation anomalies that can be compared across locations with different climatology and highly non-normal precipitation distributions (Stagge et al., 2014). SPEI is a more recent drought

index that measures the normalized anomalies in the climatic water balance, defined as precipitation minus potential evapotranspiration (PET; Vicente-Serrano et al., 2010). As opposed to SPI, SPEI takes into account atmospheric variables other than precipitation that may affect drought. Which additional atmospheric variables that are included depend on the equation chosen to estimate PET. The Hargreaves equation (Hargreaves and Samani, 1985) was used in this study following the recommendation by Stagge et al. (2014). The Hargreaves equation estimates daily PET based on each day's mean temperature, the

difference between daily minimum and maximum temperature (proxy for net radiation), and an estimate of (extraterrestrial) radiation based on the latitude and day of the year.

SPI3 (SPEI3) was computed by: 1) fitting three-months accumulated precipitation (or P-PET) in the reference period 1971–2000 to a parametric distribution, 2) transforming non-exceedance probabilities from the parametric distribution to the standard normal distribution, and finally 3) using the normal distribution to estimate the 2018 anomaly in terms of standard deviations

(Lloyd-Hughes and Saunders, 2002; Guttman, 1999; McKee et al., 1993). Both SPI and SPEI rely on the choice of a parametric distribution. This study follows the recommendations by Stagge et al. (2015), i.e. it uses the gamma distribution for the SPI calculation, incuding a "centre of mass" adjustment for zero precipitation periods, and the generalized extreme value distribution for the SPEI calculation. Except for differences in input data and transformation procedure to the standard normal distribution, the computation routine is the same for SPEI and SPI, and the multi-temporal nature and statistical interpretability of the two





indices are therefore also the same (Stagge et al., 2014). SPI and SPEI were calculated using the R-package *SCI* developed by Gudmundsson and Stagge (2016).

Dry conditions are represented by negative SPI and SPEI values, and wet conditions by positive values. A categorization of SPI values is found in Lloyd-Hughes and Saunders (2002), defining SPI absolute values of 1–1.5 (9.2 % probability) as

moderate drought/moderately wet, SPI absolute values of 1.5–2 (4.4 % probability) as severe drought/severely wet, and SPI absolute values>2 (2.3 % probability) as extreme drought/extremely wet. This categorization was used for the interpretation of the SPI3 and SPEI3 results in this study.

### 3.3   Hydrological Drought

The extremeness in streamflow and groundwater level was analysed by calculating the monthly means and ranking the low

end (low streamflow and low groundwater tables) for each month May–August in 2018, following the same procedure as for temperature and precipitation (Sect. 3.1). For streamflow, the 60 year period 1959–2018 was used as a basis for the ranking, and thus the same percentile equivalents as for temperature and precipitation apply. A 30 year period was used for groundwater due to the generally shorter time series. In this case, a rank in groundwater of 3 or 6, correspond to the 10th or 20th percentile, respectively.

The response in groundwater to climatic input is often delayed and smoothed, however, the delay may vary greatly from site to site affecting the occurrence and duration of groundwater drought (Van Loon, 2015; Van Loon and Van Lanen, 2012). Here, the delay in groundwater response to precipitation was assessed, defined as the accumulation time of the spatially collocated precipitation yielding the highest correlation between accumulated daily precipitation and daily groundwater levels for the period 1989–2018.

### 3.4   Empirical Orthogonal Function Analysis and Composite Maps

Key patterns in large scale atmospheric circulation associated with low and high summer streamflow in the Nordic region were analysed by computing the HGT500 anomalies for the years of high and low anomalies. The anomalies were identified by the three first principle components resulting from an empirical orthogonal function (EOF) analysis of the summer streamflow data. An EOF analysis allows insight into the most dominant modes of variability in a complex temporally and spatially varying

dataset by decomposing the dataset into fixed spatial patterns (EOFs) with corresponding time series (principle components, PCs), that each represent a given proportion of the total variance in the dataset (Wilks, 2006).

Summer (June-July-August) streamflow averages were computed for each year 1959–2018 and the time series standardized and detrended prior to the empirical orthogonal function analysis. The EOFs and PCs were calculated using the Python library *eofs* (Dawson, 2016). For each of the principle components (PCs), years with absolute values larger than one standard devia-

tion were defined as high (positive values) and low (negative values) anomaly years. For each set, we computed "high years composite maps" and "low years composite maps" of concurrent (average summer; June-July-August) and preceding (average spring; March-April-May) HGT500 anomaly and SST anomaly. The significance of the composite maps were estimated by a two-sided standard t-test at a 5 % significance level.



## 4 Results

### 4.1 Meteorological Situation

Figure 3 shows the evolution of SST anomalies from May–August 2018 as compared to the reference period (1981–2000; all months throughout 2018 are shown in Fig. A1). Patterns of negative and positive SST anomalies are relatively stable from

May–August, characterised by one negative and two positive anomalous SST centers. The strongest negative SST anomalies were found in an area south of Greenland (50–60° N), whereas strong positive SST anomalies were found below this area, in a belt from 20–80° W at approx. 40° N. The second region of positive SST anomalies was found in the regions surrounding Europe between 0–40° E (Barents Sea, Norwegian Sea, North Sea, Baltic Sea, Balck Sea and parts of the Mediterranean Sea). The highest SST anomalies exceeded 4° C and were found in the Baltic Sea, Black Sea and northeastern Mediterranean Sea.

Positive anomalies of similar magnitude were found in the Barents Sea in July and August.

HGT500 anomalies for each month May–August 2018 as compared to the reference period (1971–2000) are shown in Fig. 3 (all months throughout 2018 are shown in Fig. A2). May 2018 was characterised by a dipole-like structure in the atmospheric circulation, with HGT500 anomalies ranging from -120 m to 120 m. A high-pressure system (anticyclonic circulation) was centred over Fennoscandia, whereas Greenland and eastern Canada were under the influence of a low-pressure system (cyclonic

circulation). South of the cyclonic circulation, a weaker anticyclonic circulation was found over the east coast of the US. In June, the HGT500 anomalies were generally lower than in May, with anticyclonic conditions centred over the British Isles and at similar latitudes, two cyclonic circulations, one centred over the Canadian east coast and one centred over Russia at approx. 70° E. The HGT500 anomalies in July were similar to the ones in May in their spatial patterns and anomaly magnitudes, however with a slight northward shift. In August, the high-pressure systems weakened in magnitude, with a high-pressure

system located southeast of Fennoscandia, and a low-pressure system developed over the North Atlantic between Iceland and Norway.

The anomalies of the HGT500 averaged over the period May–August, in 2018, relative to 1959–2018 (represented as standard deviations, std, from the 60-year mean) for a sequence of subdomains in Europe are shown in Fig. 4a. Each subdomain covers 20°×20°lat/lon, and they are shifted one grid cell (2.5°) in longitudinal or latitudinal direction at the time. Results for

each month May–August 2018 separately are shown in Fig. A3. Most of Europe show HGT500 values of more than 2 std, and in regions centred around Denmark (between -2.5–12.5° E and 52.5–57.5° N), HGT500 deviated more than 3 std. Figure 4b shows the aggregated May–August HGT500 time series for a selected subdomain centred over Scandinavia (Scandinavian subdomain: 52.5–72.5° N and 5–25° E), demonstrating the record breaking high-pressure system averaged over the period May-August for this subdomain. As shown in Fig. A3, particular high anomalies are seen in May and June, whereas more

normal values are found in June and August. In May 2018, the std is twice as high as the second most extreme year (1993) and more than 3 std away from the mean.

Figure 5a–d shows the ranks of each month May–August 2018 (maximum) temperatures (all months throughout 2018 are shown in Fig. A4). Temperatures during this period were exceptionally high (rank 1–6), with record-breaking (rank 1) or near-record-breaking (rank 2–6) temperatures in several European regions. The most widespread extreme temperatures





were found in May, when the top six ranks (dominated by rank 1 and 2) covered almost the whole of the Nordic region and large parts of northern and eastern Europe. Record-breaking weather was reported by meteorological offices in most of the affected countries. In Norway and Germany, for example, the meteorological institutes reported that the national May temperature was the highest on (the more than 100-year) record, and 97 meteorological stations in Norway (with record

lengths between 15 and 155 years) registered record-breaking May temperature (Grinde et al., 2018b; Deutscher Wetterdienst, 2018). In June, the area covered by exceptionally high temperatures decreased, mainly covering a smaller region from northern France to Poland, southern Scandinavia and the British Isles. Ireland stands here out with record-breaking temperatures. Only southern Fennoscandia ranked 1–6 in June, however, this changed drastically in July, when almost the whole of Fennoscandia experienced the highest or second highest temperatures on the record for this month. In Norway, 43 measuring stations broke

their mean July temperature record (Grinde et al., 2018c). High ranks are also seen in regions facing the North Sea and the Baltic Sea. A southern shift is seen in August, where a southwest-eastern belt of exceptionally high temperatures extended from the Iberian Peninsula to southeastern Fennoscandia. Regions, mainly in Spain, Portugal and Germany, experienced record-breaking temperatures this month.

Record-breaking or near-record-breaking low precipitation for each month May–August (Fig. 5e–h; all months are shown

in Fig A5) were much less common and only found in smaller and more scattered areas across northeastern Europe. Some, however more extreme, clusters were found in June mainly located in southern UK, Benelux, Germany and Belarus. In July, larger clusters are seen in most of Benelux, Denmark, Fennoscandia and Germany. A relatively large region north of the Black Sea, including Moldova and parts of Romania, Ukraine and Russia, experienced record-breaking and near-record-breaking low precipitation in August. In addition, smaller clusters of exceptional low August precipitation were found in central Europe.

**4.2   Meteorological Drought**

SPI3 and SPEI3 for each month May–August 2018 are shown in Fig. 6 (all months are shown in Fig A6 for SPI3 and Fig A7 for SPEI3). SPI3 indicates moderate meteorological drought (SPI3<-1) in parts of Europe north of 45° N in May, with a few scattered areas of severe meteorological drought (SPI3<-1.5). The situation worsened to peak in July when 17 % of the grid cells had SPI3<-1.5. The most extreme meteorological drought in northern Europe (SPI3<-2) was found in July in a

region surrounding Denmark, including southern Norway, Sweden, Benelux and Germany. Regions within the British Isles and the Baltic countries also recorded extreme meteorological drought this month. In August, extreme conditions persisted in Germany and neighboring countries, whereas the meteorological drought in Fennoscandia, the Baltic countries and the British Isles generally lessened (or ceased) as compared to July. SPI3 also revealed extreme wet conditions (SPI3>2) on the fringe of the drought affected area, i.e. along the coastal regions in northern Norway and southern parts of Europe, notable the Iberian

Peninsula in May and southeastern Europe in July and August. The SPEI3 shows a similar spatial pattern as SPI3, although somewhat higher anomalies are seen at the start of the period, i.e. in May and June, for SPEI3, with 10 %, respectively 15 % of the grid cells in severe or extreme drought (i.e. values<-1.5) as compared to 5 % (May) and 11 % (June) for SPI3.





## 4.3 Hydrological Drought

The 60-year ranking of monthly lowest streamflow in 2018 in the Nordic region (Norway, Sweden, Finland and Denmark) revealed record-breaking or near-record-breaking low streamflow in several regions from June, peaking in July, and persisting in southeastern area of the region in August (Fig. 7a–d, all months in 2018 are shown in Fig. A8). In May, only two (3 %)

of the stations experienced extremely low streamflow (rank of 1–6). In June, however, 46 % of the stations had extremely low streamflow, and 13 % were record-breaking. The proportion of stations with extremely low streamflow expanded to 68 % in July (28 % were record-breaking). Extreme conditions persisted in eastern Denmark, southeastern Sweden and southern Finland throughout 2018.

The 30-year ranking of monthly lowest groundwater levels in Sweden and Norway for each month May–August 2018 are

shown in Fig. 7e–h (all months are shown in Fig A9). Four (7 %) of the stations in Norway and Sweden had extremely low groundwater levels (rank of 1–6) in May 2018. In June, 43 % of the stations had a rank of 1–6 (7 % were record-breaking), expanding to 55 % (14 % record-breaking) in July and 63 % (14 % record-breaking) in August. Ranks between 1 and 6 are seen in 38–54 % of the wells until the end of 2018. Extremely low groundwater levels did not show any distinct spatial patterns. In several cases, stations located close to each other (pies of the same point) showed different results, reflecting the importance of

local conditions in determining the groundwater level.

The delay in groundwater response to precipitation varies among the study sites from 30 to 1500 days (Fig. 8a), whereas mean groundwater levels below surface ranged from 0.36–13.4 m (median of 2.16 m; Fig. 8b). With one exception, the most extreme groundwater levels in Norway in June and July 2018 were found for the locations with the fastest response time (30–90 days). Figure 8c shows the groundwater ranks between 1 and 6 for each month throughout 2018, plotted with the response

delay along the x-axis and the mean groundwater level depth (Fig. 8b) along the y-axis. Extreme groundwater levels emerged in June in the most shallow wells (less than 3 meters depth from surface), followed by deeper wells with response delays of up to 400 days in July–August. In September, the most shallow wells with the fastest response showed less extreme ranks, whereas deeper and more slowly responding wells started to experience extreme conditions. This pattern continued throughout 2018.

## 25 4.4 Relation between Summer Streamflow and Large-Scale Atmospheric Circulation

The three first principle components of the EOF analysis explain 52 % of the detrended and standardized summer streamflow variability over the period 1959 - 2018, and their time series and loadings are shown in Fig. 9. Note that for negative EOF loadings, the corresponding PC time series is relevant with the opposite sign. The larger the absolute value of an EOF loading, the more important is the corresponding PC time series in explaining the summer streamflow behavior of a given station. EOF1

explains 23 % of the variability and is most relevant for the streamflow in the western and northern part of Norway. EOF1 is also relevant for some stations in Denmark, which are characterised by high flow when stations in Norway have low flow and vice versa. In summer 2018, PC1 was close to one standard deviation higher than the time-series average, reflecting dry conditions in western and northern Norway. Similar to EOF1, EOF2 explains 21 % of the summer streamflow variability. EOF2 is mostly





relevant for the streamflow in Denmark, southeastern Norway and southwestern Sweden. The PC2 time series indicate extreme low flow conditions in summer 2018 in these regions. A smaller amount of variability (8 %) is explained by EOF3. EOF3 reflects opposite summer streamflow conditions in the west (Norway and Denmark) relative to the east (easternmost Norway, Sweden and Finland). The PC3 value for 2018 is close to the time-series average, and thus the conditions represented by EOF3

and PC3 are not relevant for the summer 2018.

Summers of low and high streamflow were related to the prevailing large-scale atmospheric circulation by extracting the summer HGT500 of high and low years from the three first PCs time series from the summer streamflow EOF analysis. Years with absolute PC values larger than one standard deviation from the times series average were defined as high (positive values) and low (negative values) years. Summer (June–August) HGT500 composites for these years along with wind directions and

significance, are shown in Fig. 10.

Summer low flow in western and northern Norway as indicated by high PC1 values are associated with a high-pressure system centred over the Norwegian Sea and covering most of Fennoscandia and a low-pressure system centred over the British Isles and over Russia at approx. 60° E. In summers with low PC1 values, western and northern parts of Fennoscandia lies on the border between a low-pressure system in the north and a high-pressure system in the south. The years of high (low) PC2

values are associated with a low-pressure (high-pressure) system over the North Sea, flanked by a high-pressure (low-pressure) system on the central part of the north Atlantic basin and over Russia. These pressure systems cover the region with the largest EOF2 loadings, with summer high flow associated with cyclonic circulation, and summer low flow associated with the an anti-cyclonic circulation over the region. A high-pressure system over Scandinavia and a low-pressure system over Russia at approx. 40° E are observed for summers of high PC3 values, and a low-pressure system over the North Sea and southern

Scandinavia is typical for summers with low PC3 values.

## 5  Discussion

The 2018 extreme drought centred in northern Europe substantially affected the Nordic region, particularly in late spring and summer before moving southwards in August. The Nordic region has widely different hydroclimatological and terrestrial characteristics as compared to the more commonly affected drought regions of southern and central Europe. This makes the drought

of 2018 and its propagation in the hydrological cycle unique. Special for the region is a high diversity in hydroclimatological conditions, including the effect of snow on hydrology. Accordingly, the response to a meteorological drought and its propagation in the hydrological cycle will vary. Here, we discuss the 2018 drought, first from a climatological perspective (Sect. 5.1), then by the hydrological perspective (Sect. 5.2). Further, the results of the EOF analysis, linking atmospheric circulation and low summer streamflow in the Nordic region, are discussed (Sect. 5.3), followed by some final remarks on the representability

of the hydrological data used in the study (Sect. 5.4).





## 5.1 The 2018 drought from a climatological perspective

The 2018 drought confirms the central role of anticyclones in the development of northern (>40° N) Eurasian droughts (Schubert et al., 2014). The strongest HGT500 anomalies over the period May to August were found in May and July. May was characterised by a cyclonic circulation centred over Greenland and western Russia, and pronounced anticyclonic circulation

centred over the continental Nordic region extending down to central North-Atlantic and the east-coast of North America. This wave train pattern resembles the atmospheric circulation associated with the leading mode of drought variability over Europe as presented by Ionita et al. (2015). Large parts of the region experiencing anticyclonic conditions in the months from May to August, also showed extreme temperatures (defined as having a rank between 1 and 6). The stronger the HGT500 anomaly, the more extreme the temperature, emphasising the strong link between the two variables.

Overall, the observed positive SST anomalies in summer 2018 overlap with the anticyclonic circulations (positive HGT500 anomalies) in May and July 2018. Anomalous anticyclonic circulation, as observed in these two months, may decrease convection and increase incoming solar radiation, leading to warmer SST in the underlying seas (Feudale and Shukla, 2011). The spatial pattern of SST anomalies in 2018 are similar to those in the summers of 2003 and 2015, representing two of the most extreme drought events in Europe in recent years (Ionita et al., 2017; Laaha et al., 2016; Black et al., 2004). During all three

events, a persistent negative anomaly was centred south of Greenland May–August. The anticyclonic centres and associated temperature extremes over continental Europe in 2018, were generally located more towards the northeast as compared to the 2003 and 2015 events. An overlapping region in central Europe experienced temperature extremes all three summers (Ionita et al., 2017; Fischer et al., 2007b). Overall, most major European streamflow droughts between 1960–90 were associated with high-pressure systems across central Europe (Stahl, 2001), highlighting the unique location of the 2018 event. This is especially

the case for May and July, in which the high-pressure system centred over the Nordic region is more than 3 std, respectively 2 std, away from the 60-year mean (Fig. A3). However, in August 2018, the hot temperature extremes covered a region extending from the southwest to northeast Europe, resembling the affected region in summer of 2015 (and to a lesser degree 2003).

Monthly precipitation extremes across the period May–August were not as widespread as temperature extremes, however, areas with extreme low precipitation (rank between 1 and 6) generally also experienced extreme high temperatures. Overall,

the region affected was located further north as compared to previous large-scale droughts in Europe, such as the summer droughts in 2003 and 2015 (Ionita et al., 2017). The SPI3 and SPEI3 both showed a similar northern European located drought, however, as both these indices reflect a 3-month accumulated deficit in precipitation, respectively a climatic water deficit, a higher consistency is seen in time. Furthermore, both indices show dry conditions already in May, reflecting the conditions in the months March–May. As seen in Fig. A4, extreme high temperatures are seen already in April in parts of the region (central

Europe), potentially leading to drier than normal conditions in the soils. For both SPI3 and SPEI3, the drought peaks in July.

Overall, the percentage of grid cells showing extreme drought is higher for SPEI3, highlighting the importance of looking not only at precipitation when analysing the impact of drought, as also previously recognized when comparing the SPI and SPEI for Europe (e.g. Stagge et al., 2017). The use of potential evapotranspiration in SPEI (rather than actual evapotranspiration) might be less an issue in the Nordic region, where evapotranspiration in general is limited by energy, as opposed to water-limited areas





dominating in central and southern Europe (McVicar et al., 2012). The inclusion of potential evapotranspiration in SPEI (as opposed to using only precipitation in SPI), might therefore prove suitable for drought assessments in energy-limited regions. However, water may become a limiting factor also in these regions in exceptional years, such as the summer of 2018. As the soil dries out, it may give rise to a positive land-atmosphere feedback, i.e. an enhanced warming is seen as less energy is spent

on evapotranspiration. Such soil moisture-temperature feedbacks have played an important role in the evolution of previous European heat waves (Fischer et al., 2007a), and may have played an important role in the 2018 event as well. Being outside the scope of this study, this would be an interesting aspect of a further study.

## 5.2 The 2018 drought from a hydrological perspective

Overall, drought impacts are related to deficits in different components of the hydrological cycle, not in the meteorological

variables as such. Key impacts of the 2018 drought were related to soil moisture (crop failure and wild fires) and hydrological drought (e.g. impacts on energy, water supply and aquatic ecosystems). As the drought propagate, the event is normally lagged, attenuated and lengthened as compared to the original meteorological event (Van Loon and Van Lanen, 2012; Van Loon et al., 2011), the question being to what degree, which will vary with event and region impacted. Furthermore, antecedent water storage (initial conditions), such as snow, glaciers and groundwater, play an important role in the occurrence, timing and

development of the hydrological drought.

In regions affected by seasonal snow, drought occurrence and propagation is to a large degree influenced by the snow storage and snowmelt timing as compared to a normal year. During the snow accumulation season in 2018, above normal precipitation fell in early winter in most of the Nordic region, and less than normal precipitation in western/northern Norway and Finland towards the end of the snow season (as indicated by SPI3; Fig. A6a–c). Most of the snow-dominated catchments

(with the exception of the northernmost part of the Nordic region), experienced meteorological drought in May–July. Record high temperatures emerging during the snowmelt season (i.e. in May), and 19 stations (24 %), all with a mountain or inland regime, experienced one of their six highest May streamflow since 1959. For the other stations affected by snowmelt, however, a more normal flood situation followed, one hypothesis being that part of the snow was lost due to sublimation. In addition, higher than normal evapotranspiration rates led to less water feeding the streams. The high snowmelt and evapotranspiration rates

likely caused an earlier end of the snowmelt season as well as a smaller total volume of melt water contribution to streamflow compared to normal (given the same preconditions). Following the snowmelt peak, streamflow drought started emerging in June in large parts of the Nordic region. Noteworthy exceptions are the three glacier dominated streamflow stations, for which high summer temperatures led to high melt rates and sustained water contribution from the glaciers.

Streamflow stations without a snow season are mainly located in Denmark and southern Sweden. Denmark did not expe-

rience a meteorological drought until July, despite extreme temperatures in May. Accordingly, streamflow drought was first observed in July and (to a lesser degree) August, for stations located in the eastern parts of Denmark. Most of the stations in western Denmark, however, did not experience extremely low streamflow at all during May–August 2018. As a whole, Denmark had extremely low precipitation and severe to extreme meteorological drought, as indicated by SPI3, in July. The spatial pattern of extreme temperature (and SPEI3) this month, however, reflects the east-west deviation in extremely low streamflow



in Denmark, indicating that higher than usual evapotranspiration rates likely has contributed to extreme conditions in the east. In the west, on the other hand, persisted groundwater contribution to streamflow combined with less extreme evapotranspiration losses may have prevented streamflow drought to develop.

Whereas exceptional conditions sustained in the southeastern area of the Nordic region (eastern Denmark, southeastern Sweden and southern Finland) throughout 2018, streamflow in the north and western part of the region was replenished by heavy precipitation in August (Grinde et al., 2018a). This divide reflects the southeastern movement of the anticyclonic circulation as well as the cyclonic circulation over the Norwegian Sea in August, and winds moving northeast from the North Sea, bringing precipitation towards the coast along with them. The precipitation did not only replenish the rivers, but led to extremely wet conditions at several streamflow stations. Western and northern stations experienced one of their six highest monthly streamflow since 1959 in August (5 stations), September (21 stations) and October (16 stations). Accordingly, the streamflow drought ends in August in western and northern part of the Nordic region, whereas in the southeastern area, extreme conditions persisted for several stations towards the end of the 2018.

The groundwater wells were all located in the area affected by moderate to extreme meteorological drought, as indicated by SPEI3, in May, June, July and (to a lesser degree) August. However, a high local variability is seen for groundwater drought (rank between 1 and 6), reflecting neither the spatial pattern of meteorological drought, extremely low streamflow, nor the span in groundwater regimes. The high spatial variability in hydrogeological properties across the Nordic region is mirrored in the diversity in groundwater response to meteorological conditions. Except for four wells that experienced low groundwater levels already from March, no wells showed groundwater drought in May. Similar to streamflow, this is likely due to wet preconditions, such as high groundwater levels and/or snow volumes recharging groundwater during the melt season. In June, extreme conditions are found among the most shallow groundwater wells, probably due to high evapotranspiration rates in combination with precipitation deficits. From July onwards, extreme conditions are found in wells of increasing depth and response time. The extreme conditions started to cease in the shallowest and fast responding wells from September. At the end of year, 38 % of the wells still experienced extreme conditions, and below normal groundwater levels persisted well into 2019 (e.g. Table 1,n).

## 5.3 Atmospheric Circulation Associated with Low Summer Streamflow in the Nordic Region

The EOF analysis revealed that more than half (52 %) of the variability in summer streamflow, over the period 1959–2018, in the Nordic regions, can be explained by the three first principle components, whereof the two first EOFs explain 44 % (Fig. 9). The analysis is somewhat biased towards Danish conditions, as the station density is much higher here compared to the rest of the region, in particular Sweden and Finland. EOF1 and EOF2 indicate a division in summer streamflow variability between western/northern Norway and the southeastern part of the Nordic region. During 1959–2018, only two summers had summer low flow in the whole region (i.e. 1969 and 2006). These two summers are also previously identified as dry by different drought indices (e.g. Spinoni et al., 2015; Hannaford et al., 2011), and are years with overlying May–August HGT500 anomalies of more than 1 std above the 1959–2018 average (Fig. 4).





High values of summer PC1 indicate low summer streamflow in the northwestern part of the Nordic region, and are associated with a high pressure system over the Norwegian Sea (Fig. 10). Several of the streamflow stations with strong EOF1 loadings recorded extremely low streamflow values in June and July 2018. However, the summer of 2018 was not a high anomaly year in PC1, which might be due to the heavy precipitation in August 2018 replenishing the rivers in this region.

Low values of PC2 indicate low summer streamflow in the southeastern part of the Nordic region, with the summer of 2018 as the most extreme year. The main reason for this might be the extreme conditions throughout June–August at several of the stations that have the strongest EOF loadings along the Sweden-Norway border and southern Sweden (Fig.7b–d). After 2018, the most extreme low years, as indicated by PC2, are 1975–76. This period has previously been identified as benchmark drought event in western and northern Europe (e.g. Zaidman and Rees, 2000; Stahl, 2001). Low values of PC2 are also associated with

a high-pressure system over the North Sea, surrounded by low-pressure systems over Greenland/north Atlantic, Russia and the Mediterranean region. The pattern has some resemblance with the Scandinavian teleconnection pattern (SCAN). Interestingly, May 2018 has the highest May SCAN value (of 1.69) and July 2018 the third highest July SCAN value (of 2.27, the highest being 2.61 from 1997) over the period 1950–2019 (data from https://climexp.knmi.nl/data/icpc_sca.dat, retrieved 14.04.2020).

### 5.4  Hydrological data representatitivity

The streamflow dataset used in this study covers a rather wide range of catchments areas (6.6–10864 km$^2$), and includes stations across all of Norway, Denmark, Sweden and Finland. However, the density of stations varies, being much higher in Denmark and Norway as compared to Sweden and Finland. This lack of spatial representation affects the EOF analysis in particular, but also the percentages of stations with extremely low streamflow.

The number of wells included in the groundwater dataset was strongly limited by the requirement of no/little human influence

(or lack of knowledge thereof), data quality and the period defined. The selected groundwater wells are relatively shallow, with a median depth of 2.16 m below surface, however, the range across the region or average value is not known, thus it is difficult to state whether this is a representative set of wells or not. Nevertheless, the large span in the 'delay in groundwater response' variable suggests a good coverage. The groundwater dataset only covers Sweden and (southern) Norway, and as much as 46 of the 56 stations are located in Sweden, thus the results would be biased towards Swedish conditions. In addition, several of

the well are located at the same site often at different depth, affecting the spatial representatitivity of the dataset. However, these wells highlight the high local variability seen in the groundwater level response reflecting the local heterogeneity in hydrogeological properties, and thus cautions the local relevance of conclusions regarding groundwater drought made at the regional scale.

### 6  Conclusions

This study characterised the 2018 northern European drought from both a climatological and hydrological perspective. This event was unique in its northern location, affecting a region with highly diverse hydrometeorological conditions compared to the more central and southern parts of Europe, recently hit by major droughts such as the events of 2003 and 2015.



The North Atlantic Ocean and seas surrounding Europe experienced persistent anomalously high SST from May–August and record-breaking temperatures over the Nordic region in May and July, associated with record-breaking high-pressure systems overlying the region. Extreme monthly precipitation deficits were not as wide spread as the extreme monthly temperatures, however the persistent lack of precipitation from May–July led to extreme meteorological drought (estimated by SPI3) in a

region surrounding Denmark, including southern Norway, Sweden, Benelux and Germany. The meteorological drought in this region was considered even more extreme when considering the climatic water balance (precipitation minus potential evapotranspiration) using the SPEI3 index, emphasising the importance of accounting for temperature (and not solely precipitation as in SPI) in meteorological drought assessments. After July, the high-pressure system shifted southward, centred in Germany, and meteorological drought was only seen in small clusters across the Nordic region.

Whereas record-breaking temperatures and moderate meteorological drought emerged over most of the Nordic region in May, hydrological drought (estimated as monthly ranks of streamflow and groundwater) did not appear before June. The effect of snow is an important hydrological characteristic over large parts of the region, and at many locations the streamflow were still fed by meltwater during May 2018. The number of stations experiencing extremely low streamflow (rank between 1 and 6) expanded from 43 % in June to 68 % in July. Stations with more than 30 % of their catchment covered by glaciers

did not experienced streamflow drought during the summer due to the continuous contribution of glacial melt water. In mid-August, heavy precipitation replenished rivers in western and northern parts of the Nordic region, whereas extremely low streamflow persisted throughout 2018 in the southeastern parts. Groundwater drought peaked in August with 63 % of the stations experiencing extremely low groundwater levels (rank between 1 and 6). The spatial pattern of groundwater drought as it developed was heterogeneous, and an interpretation of the patterns only made sense when looking at the groundwater depth

and 'response delay to precipitation' combined. Extremely low groundwater levels emerged in the shallowest wells in June. With time, extreme conditions were found in wells of increasing depth and response delay, and by the end of 2018, 38 % of the wells still had extreme low groundwater levels. The high local variability observed in the development of groundwater drought in 2018, highlights the care and awareness needed when analysing groundwater drought at the regional scale based on local well data that varies in depth and site characteristics.

The leading modes of Nordic summer streamflow variability 1959–2018 revealed a distinction in summer streamflow variability between the western/northern part and the southeastern part of the region. As identified by composite maps of summer geopotential height anomalies, high-pressure systems centred over the Norwegian Sea and the North Sea were associated with low summer streamflow in the western/northern and southeastern part of the Nordic region, respectively. In both cases, significant high-pressure systems overlay the region experiencing low summer streamflow, emphasising the important link between

streamflow variability and large-scale atmospheric circulation.

The complexity of the 2018 drought event as revealed by the large variability in drought characteristics seen across space and time in the Nordic region, serves as yet another example of the care needed when analysing drought in different components of the hydrological cycle. The diversity, caused by high local variability in terrestrial properties, implies a different response to the meteorological forcing and thus, different footprints of meteorological and hydrological drought. As the majority of drought





impacts are felt on the ground, and thus more directly related to hydrology than meteorology, it is important to incorporate variables other than weather alone, when characterising drought.

*Data availability.* Our study is based on third party data. Citations to the data sets are included in the reference list, and data providers are acknowledged in the acknowledgment section.

5 *Author contributions.* SJB, MI and LMT designed the study. MI performed the analysis and visualization of the geopotential height anomalies, and SJB performed the analysis and visualization of the remaining. SJB and LMT prepared the original draft. All authors reviewed and edited the final manuscript.

*Competing interests.* The authors declare that they have no conflict of interest.

*Acknowledgements.* We acknowledge the E-OBS dataset from the EU-FP6 project UERRA (http://www.uerra.eu) and the Copernicus Cli-
10 mate Change Service, and the data providers in the ECA&D project (https://www.ecad.eu) NOAA High Resolution SST data was provided by the NOAA/OAR/ESRL PSD, Boulder, Colorado, USA, from their Web site at https://www.esrl.noaa.gov/psd/. We thank the Norwegian Water Resources and Energy Directorate (NVE), Danish Environment Portal for Denmark, Swedish Meteorological and Hydrological Institute (SMHI) and Finnish Environmental Institute (SYKE) for providing streamflow data for Norway, Denmark, Sweden and Finland, respectively. We also thank NVE and the Geological Survey of Sweden (SGU) for providing groundwater data for Norway and Sweden,
15 respectively. Funding by the AWI Strategy Fund Project - PalEX and by the Helmholtz Climate Initiative - REKLIM are gratefully acknowledged. This paper supports the work of the UNESCO-IHP VIII FRIEND programme and the Panta Rhei Initiative of the International Association of Hydrological Sciences (IAHS).





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





**Table 1.** Reports and news articles about 2018 heat and drought related impacts.

| Ref. | Publisher | Region | URL (last access) | Last update |
|---|---|---|---|---|
| a | Joint Research Centre, European Commission | Europe | https://op.europa.eu/en/publication-detail/-/publication/435ef008-14db-11ea-8c1f-01aa75ed71a1/language-en (24.03.20) | 29.11.18 |
| b | Agriculture and Rural Development, European Commission | European Union | https://ec.europa.eu/info/news/drop-eu-cereal-harvest-due-summer-drought-2018-oct-03_en (24.03.20) | 31.10.19 |
| c | Reuters | European Union | https://www.reuters.com/article/us-europe-grains-analyst/analysts-cut-eu-wheat-crop-outlook-again-on-catastrophic-north-idUSKBN1KU15E (24.03.20) | 09.08.18 |
| d | Euronews | Europe | https://www.euronews.com/2018/08/10/explained-europe-s-devastating-drought-and-the-countries-worst-hit (24.03.20) | 12.08.18 |
| e | Swedish Board of Agriculture | Sweden | https://www2.jordbruksverket.se/download/18.21625ee16a16bf0cc0eed70/1555396324560/ra19_13.pdf (24.03.20) | 16.04.19 |
| f | Norwegian Agriculture Agency | Norway | https://www.landbruksdirektoratet.no/no/statistikk/landbrukserstatning/klimarelaterte-skader-og-tap/avlingssvikt-statistikk (24.03.20) | 02.09.19 |
| g | Norwegian Water Resources and Energy Directorate | Norway, Sweden, Finland | https://www.nve.no/Media/7385/q3_2018.pdf (24.03.20) | 17.10.18 |
| h | newsinenglish.no | Norway | https://www.newsinenglish.no/2018/07/13/drought-blamed-for-high-electricity-rates/ (24.03.20) | 13.07.18 |
| i | Handelsblatt Today | Germany | https://www.handelsblatt.com/today/companies/low-water-dwindling-rhine-paralyzes-shipping-transport/23695020.html?ticket=ST-3121873-yvfwqi1ee3yWiy3BK1Zv-ap4 (24.03.20) | 27.11.18 |
| j | Reuters | Hungary | https://www.reuters.com/article/us-europe-weather-hungary-shipping/water-levels-in-danube-recede-to-record-lows-hindering-shipping-in-hungary-idUSKCN1L71DH (24.03.20) | 22.08.18 |
| k | Deutsche Welle | Germany | https://www.dw.com/en/hot-weather-exposes-world-war-ii-munitions-in-german-waters/a-44924959 (24.03.20) | 02.08.18 |
| l | Business Insider | Czech Republic | https://www.businessinsider.com/sinister-hunger-stones-dire-warnings-surfaced-europe-2018-8?r=US&IR=T (24.03.20) | 27.08.18 |
| m | Adresseavisen | Norway | https://www.adressa.no/nyheter/trondelag/2018/07/28/Gaula-stengt-for-fiske-på-grunn-av-varmen-17208221.ece (24.03.20) | 30.07.18 |
| n | The Local Sweden | Sweden | https://www.thelocal.se/20190425/sweden-may-be-heading-for-a-new-water-crisis (24.03.20) | 25.04.19 |




**Table 2.** Variables, extremeness indices and spatial coverage used to characterise the 2018 meteorological situation, meteorological drought and hydrological drought. All indices are calculated on a monthly basis.

| | Variable(s) | Extremeness index | Spatial Coverage |
|---|---|---|---|
| *Meteorological situation* | | | |
| | Sea Surface Temperature (SST) | 2018 anomaly (in degree Celsius) relative to 1981–2000 | Europe and surrounding regions |
| | Geopotential Height at 500 mb (HGT500) | 2018 anomaly (in meters) relative to 1971–2000 | Europe and surrounding regions |
| | Geopotential Height at 500 mb (HGT500) | 2018 anomaly (in standard deviations from the mean) relative to 1959–2018 for European subdomains | Europe and surrounding regions |
| | Maximum Temperature (Tx) | Rank of 2018 based on highest 1959–2018 maximum temperatures | Europe |
| | Precipitation (P) | Rank of 2018 based on lowest 1959–2018 precipitation | Europe |
| *Meteorological drought* | | | |
| | Precipitation | Three-months Standardized Precipitation Index (SPI3) of 2018 relative to 1971–2000 | Europe |
| | Precipitation, and Minimum, Maximum and Mean Temperature | Three-months Standardized Precipitation Evapotranspiration Index (SPEI3) of 2018 relative to 1971–2000 | Europe |
| *Hydrological drought* | | | |
| | Streamflow | Rank of 2018 based on lowest 1959–2018 streamflow | Norway, Sweden, Finland and Denmark |
| | Groundwater | Rank of 2018 based on lowest 1989–2018 groundwater level | Norway and Sweden |





**Figure 3.** Left panel: Sea Surface Temperature (SST) anomalies for (a) May, (b) June, (c) July and (d) August 2018 relative to the reference period 1981–2000. Right panel: Geopotential Height at 500 mb (HGT500) anomalies for (e) May, (f) June, (g) July and (h) August 2018 relative to the reference period 1971–2000. Zonal and meridional wind at 500 mb level are added to indicate wind directions.

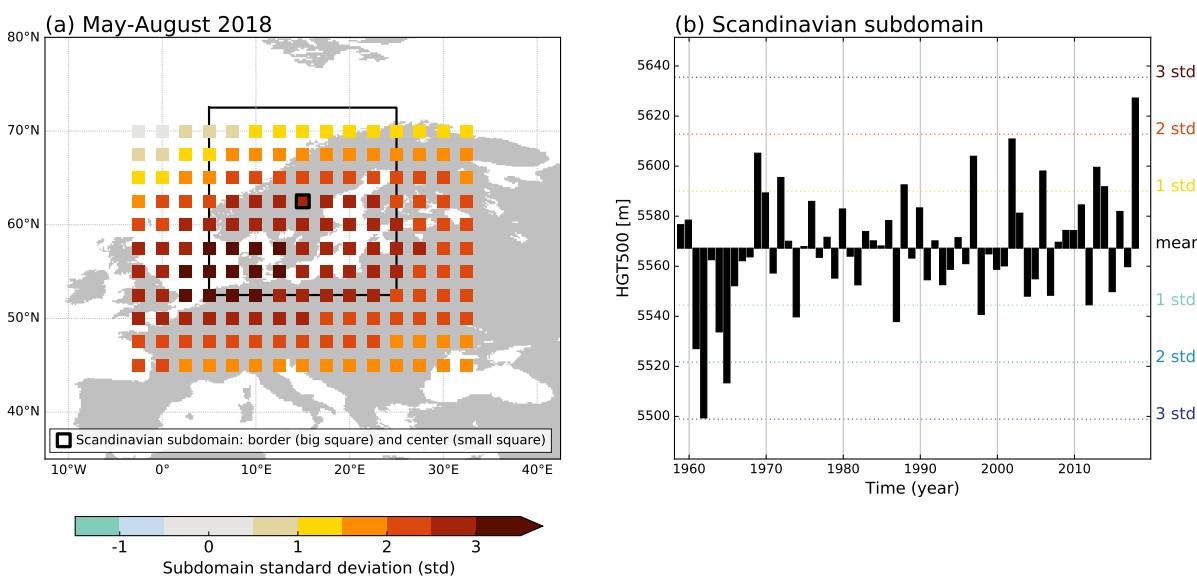

**Figure 4.** (a) Geopotential height at 500 mb (HGT500) shown as standard deviation (std) of aggregated May–August 2018 based on the 60-year period (1959–2018) for subdomains of 20° lon/lat throughout Europe, shifted 2.5° at a time. The coloured squares are the center points of each subdomain. This is illustrated for one subdomain over Scandinavia, with a large square and a small square marking the subdomain's border and centerpoint, respectively. (b) Aggregated May–August HGT500 1959–2018 time series for the Scandianvian subdomain marked in (a).

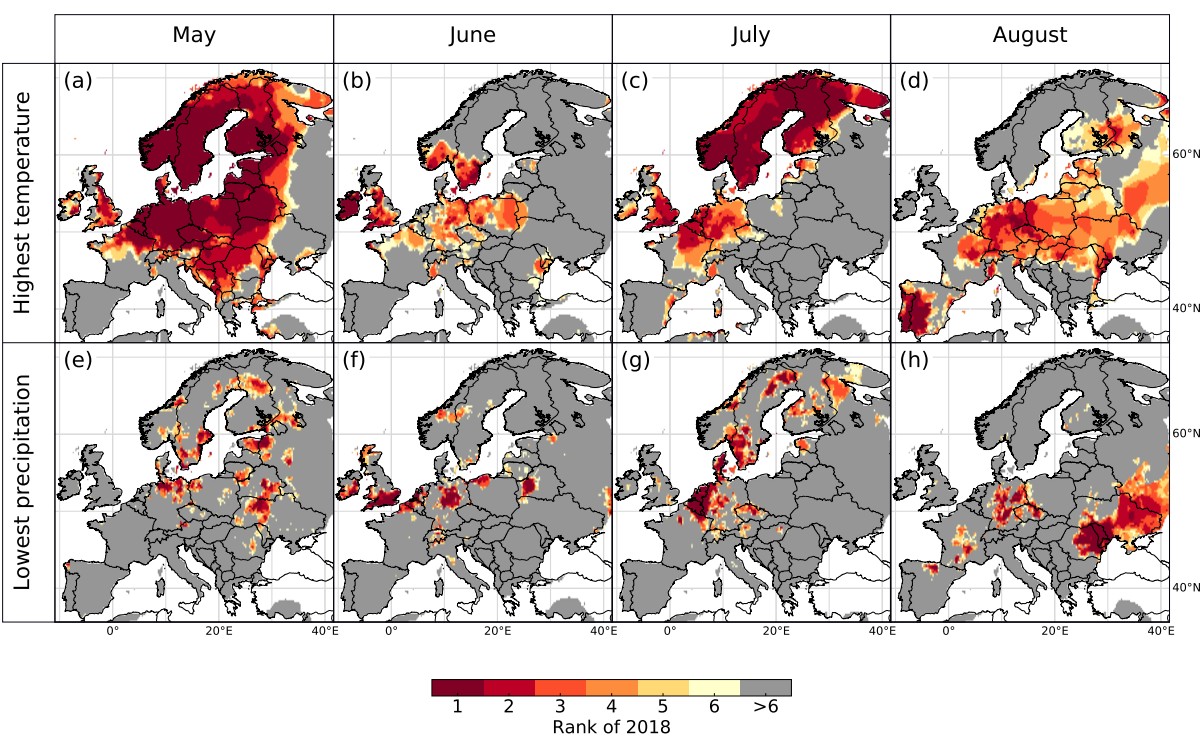

**Figure 5.** Top-six ranking of 2018 highest temperature (monthly mean of daily maximum temperature) for (a) May, (b) June, (c) July and (d) August, and top-six ranking of 2018 lowest precipitation for (e) May, (f) June, (g) July and (h) August. Analysed period is 1959–2018. A rank of one signifies that 2018 had the warmest (in the case of temperature) or driest (in the case of precipitation) month since 1959, a rank of two signifies that 2018 had the second most extreme value in that month, etc.

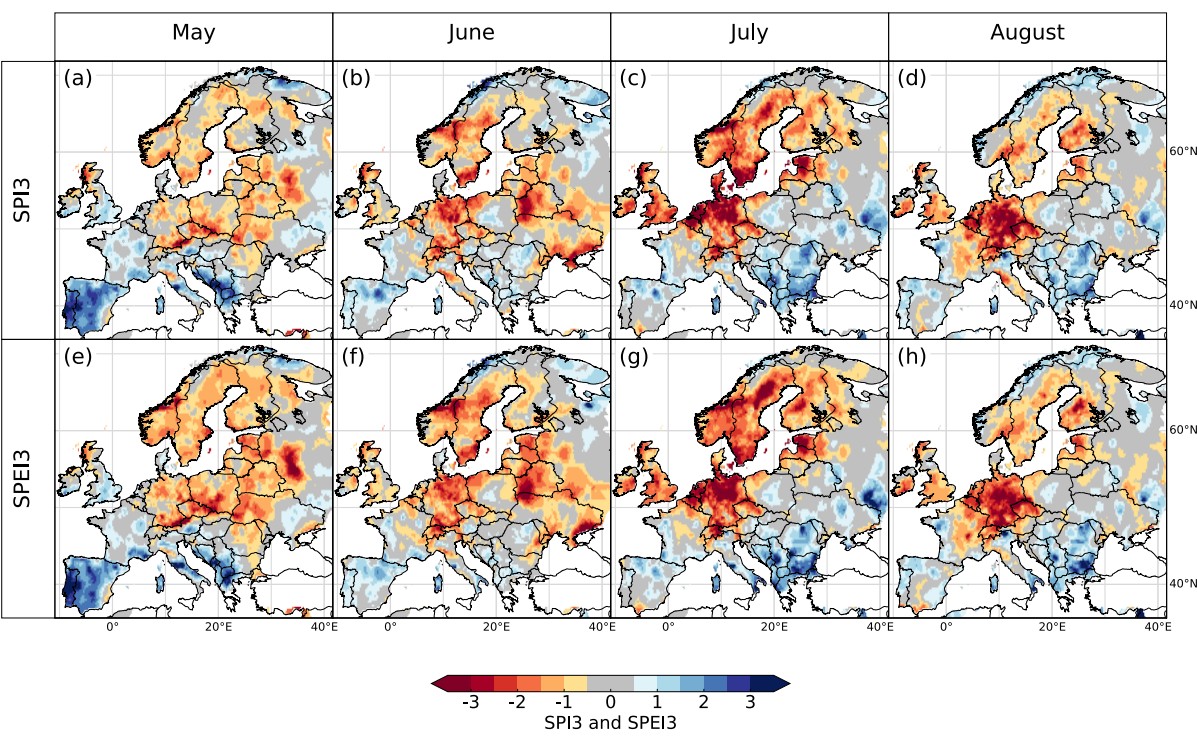

**Figure 6.** Meteorological drought 2018 indexed by SPI3 for (a) May, (b) June, (c) July and (d) August, and SPEI3 for (e) May, (f) June, (g) July and (h) August. Reference period used is 1971–2000.



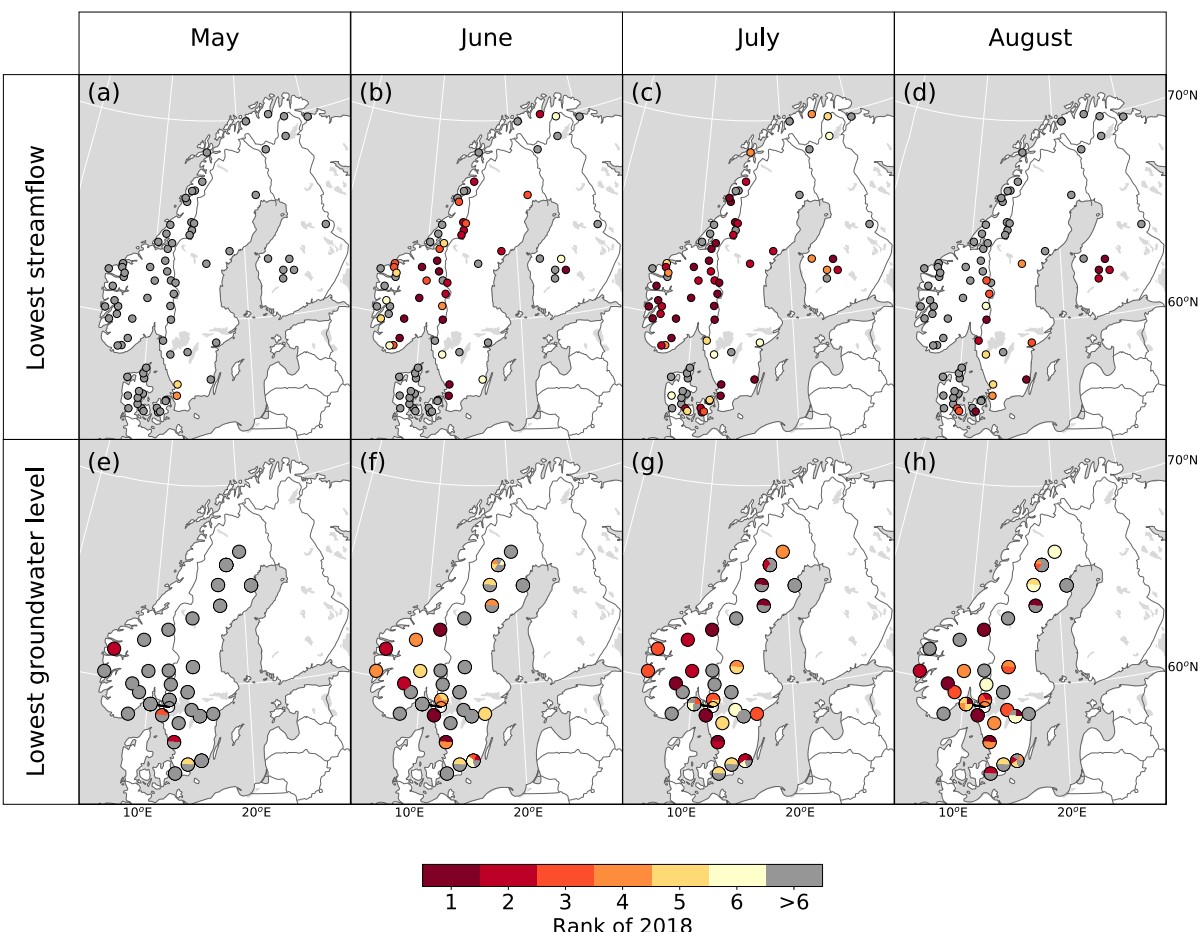

**Figure 7.** Top-six ranking of lowest streamflow for (a) May, (b) June, (c) July and (d) August, and top-six ranking of 2018 lowest groundwater level for (e) May, (f) June, (g) July and (h) August. Analysed period is 1959–2018 for streamflow and 1989–2018 for groundwater. A rank of one signifies that 2018 had the lowest monthly streamflow since 1959 (upper panel), or the lowest groundwater level in that month since 1989 (lower panel). A rank of two signifies that 2018 had the second most extreme value in that month, etc.

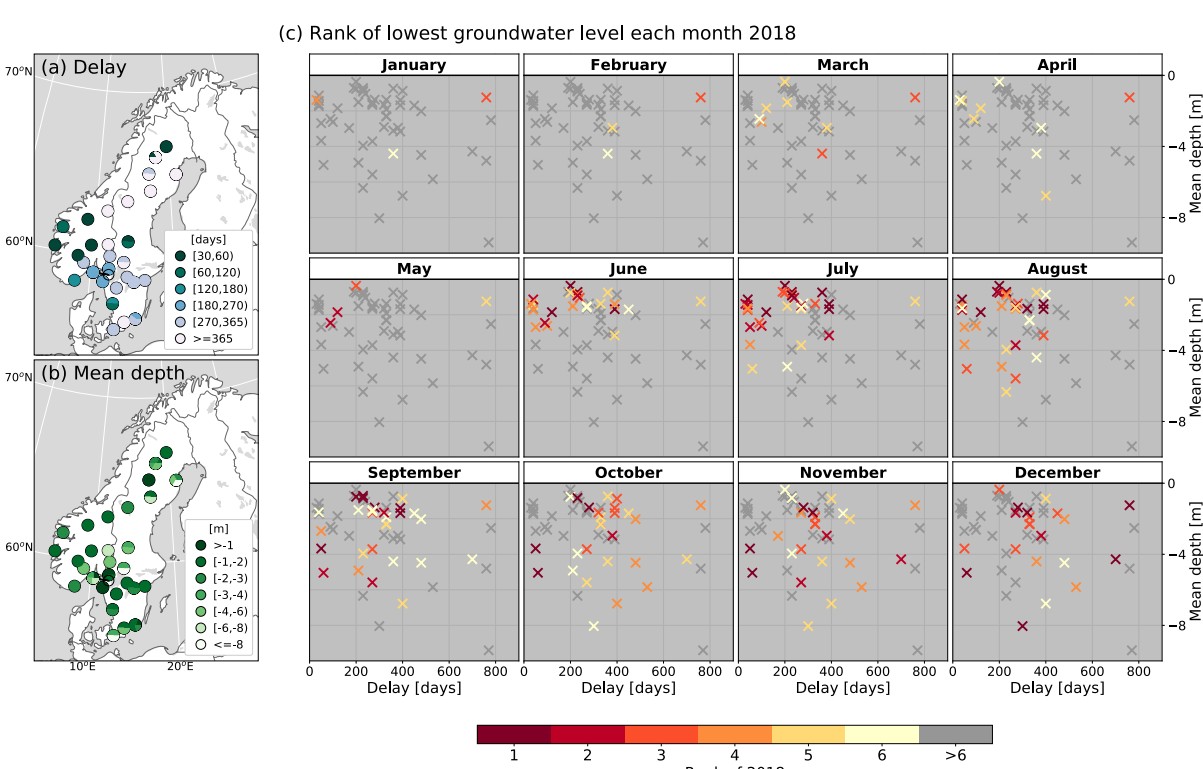

**Figure 8.** (a) Groundwater response to precipitation, (b) mean 1989–2018 groundwater depth below surface, and (c) top-six ranking of lowest groundwater level in each month of 2018 plotted with each well's delay and mean depth along the x-axis and y-axis, respectively. Two wells, one with delay of 1500 days and one with mean depth of -13.4 m, are outside the range of the ranking plots. Those two wells have no rank of 1–6 in April–December 2018.

**EOF analysis of summer streamflow**

**Figure 9.** Empirical Orthogonal Function (EOF) analysis based on aggregated summer (June–August) standardized and detrended streamflow (1959–2018). Maps (a–c) show the EOF loadings and time series (d–f) show the three first principle components (PCs). The explained variability of each mode is given in brackets in the corresponding EOF plot. For each of the PCs, years with absolute values larger than one standard deviation (std) are highlighted as high (positive values) and low (negative values) anomaly years.

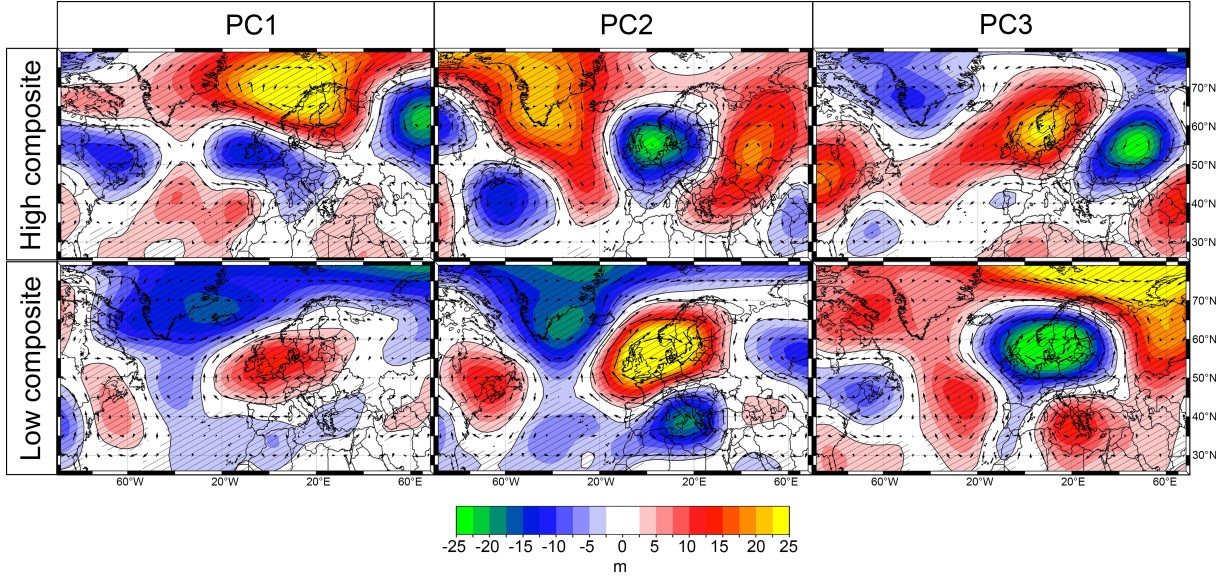

**Figure 10.** Composite maps of summer (June–August) Geopotential height at 500 mb (HGT500) anomaly relative to 1971–2000 for the first three PCs. High and low composites maps are shown, representing years with (positive and negative) values more extreme than one standard deviation in the corresponding PC time series.





**Appendix A**

**A1**

Streamflow regimes shown in Fig. 1 are calculated following the regime classification of Gottschalk et al. (1979), which consists of five main classes: Mountain, Inland, Transition, Baltic, and Atlantic regime. The classification is based on when

high and low monthly streamflow typically occur during the year. Exact periods of low and high flow occurrences for each class are not provided in Gottschalk et al. (1979). Thus, we had to make choices regarding which months to be included for the definition of each class Specifically, we calculated the 1959–2018 average streamflow for each month, and classified the stations as follows:

– Mountain regime is characterised by the two months of lowest flow occurring in winter or early spring due to snow
accumulation, and the three months of highest flow occurring in spring or early summer due to snow melt. Because the snow melt season typically occur later with increasing altitude or latitude, a somewhat generous period for snow melt was applied. Mountain regime was assigned to stations with the two lowest monthly flow in January–April and three highest monthly flow in March–August. Whereas most Mountain regimes in this study had a distinct maximum flow in May or June, three of them had a later and less distinct peak in July. The later high flow peak is explained by the
contribution of melt water from glaciers, which cover more than 30 % of the catchment of these three stations.

– Inland regime also have low flow during winter or early spring and high flow during snow melt, however, the second or third highest monthly flow occur during rainfall in autumn. Thus, the same months as for Mountain regime were used to define low flow period and snow melt period, whereas the autumn period was defined as September–November.

– Atlantic regimes have the highest monthly flow in autumn or winter due to rainfall, and the two months with lowest
flow during summer or autumn due to high evapotranspiration and/or low precipitation. Atlantic regime was assigned to stations with the highest monthly flow in September–February and the two lowest monthly flows in June–October.

– Baltic regime have the same definition of the low flow period as Atlantic regime. However, either the second or third highest monthly flow occurs in September–February, whereas the highest flow occur during the snow melt period, here defined as March–May.

– Transition regime was assigned to stations that was not assigned to any of the other regimes, and is an intermediate regime between Inland and Baltic regime.

**A2**

The groundwater regime classification in Fig. 2 is based on the classification of groundwater fluctuation patterns by Kirkhusmo (1988) who divides groundwater fluctuation patterns into three idealised regions, with the possibility of a time shifted version

of each fluctuation pattern. Region I represents groundwater levels reaching their maximum in late winter or early spring and





their minimum in late summer (similar to Atlantic streamflow regime), Region II consists of groundwater levels with two annual maxima and two annual minima (similar to Transition streamflow regime), and Region III represent groundwater levels with a minimum just before the snowmelt, and a maximum after the snowmelt (similar to Mountain streamflow regime). In this study, we calculated the 1989–2018 average groundwater level for each month and defined the classes as follows:

- If the three months with highest groundwater level occur in October–April or in January–May and the three months with lowest groundwater level occur in June–December, the groundwater station was classified as belonging to Region I.

- If the three months with highest groundwater level occur in April–July and all the three months with lowest groundwater level occur in December–May, the station was classified as Region III.

- If instead of during April–July, the three months with highest groundwater level occur in May–November, and the three
10 months with lowest groundwater level still occur in December–May, we assumed a time-lag effect and the station was classified as Region III delayed.

- If neither of the above, and the groundwater level have two minima and two maxima during the year, the groundwater station was classified as Region II.

- For the remaining three stations, the three months with highest groundwater level occurred in April–July and the three
15 months with lowest groundwater level occurred in October–January, and these stations were classified as Region I delayed.

**A3**

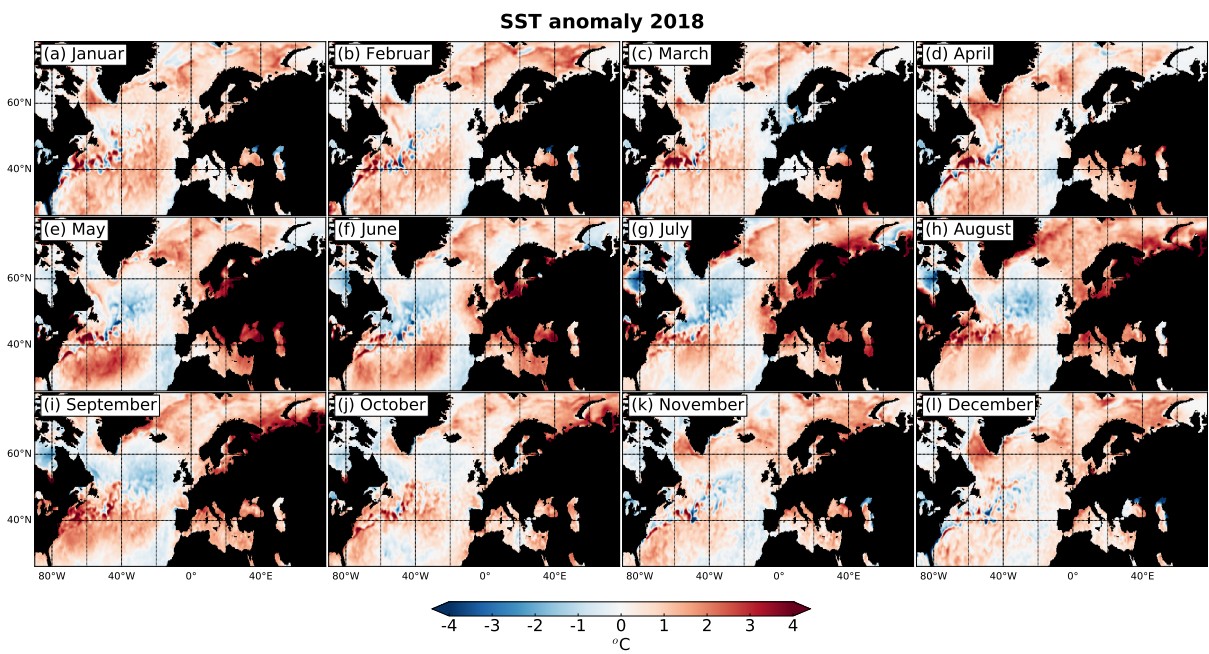

**Figure A1.** Monthly Sea Surface Temperature (SST) anomalies throughout 2018 relative to the reference period 1981–2000.

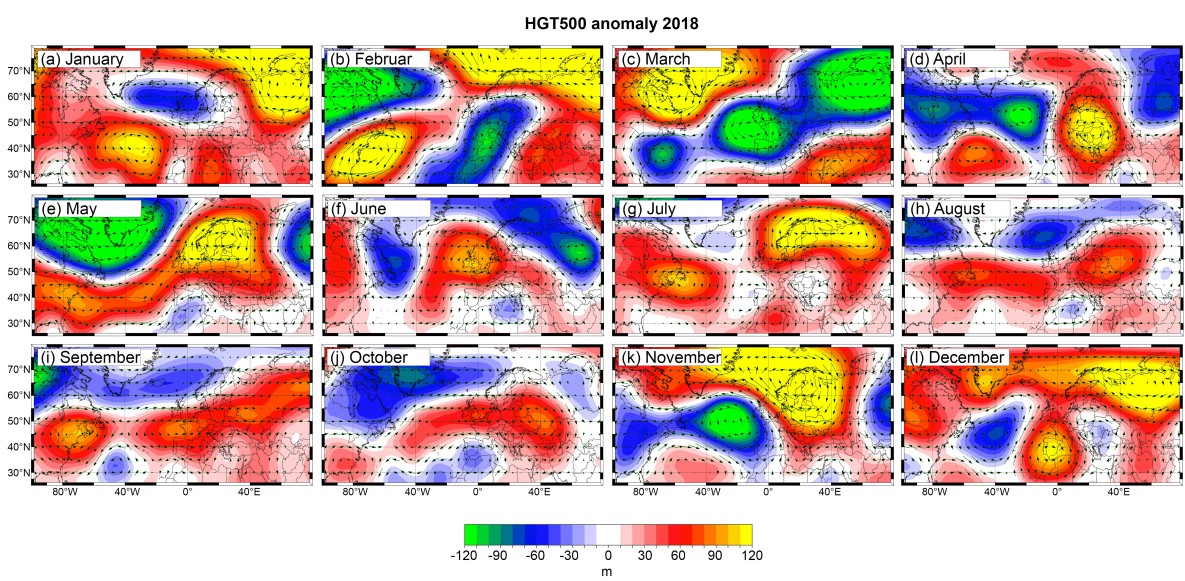

**Figure A2.** Monthly geopotential height at 500 mb (HGT500) anomalies throughout 2018 relative to the reference period 1971–2000.

**Figure A3.** Left panel: Geopotential height at 500 mb (HGT500) shown as standard deviation (std) of (a) May, (b) June, (c) July and (d) August 2018 based on the 60-year period (1959–2018) for subdomains of 20° lon/lat throughout Europe, shifted 2.5° at a time. The coloured squares are the center points of each subdomain. This is illustrated for one subdomain over Scandinavia, with a large and a small square marking the subdomain's border and centerpoint, respectively. Right panel: HGT500 1959–2018 time series of (e) May, (f) June, (g) July and (h) August for the Scandinavian subdomain.

**Figure A4.** Top-six ranking of 2018 monthly highest temperature (monthly mean of daily maximum temperature) relative to the period 1959–2018.



**Rank of lowest precipitation 2018**

Rank of 2018

1  2  3  4  5  6  >6

**Figure A5.** Top-six ranking of 2018 monthly lowest precipitation relative to the period 1959–2018.

**Figure A6.** Monthly meteorological drought indexed by SPI3 throughout 2018 relative to the reference period 1971–2000.

**SPEI3 2018**

**Figure A7.** Monthly meteorological drought indexed by SPEI3 throughout 2018 relative to the reference period 1971–2000.

**Rank of lowest streamflow 2018**



**Figure A8.** Top-six ranking of 2018 monthly lowest streamflow relative to the period 1959–2018.



**Figure A9.** Top-six ranking of 2018 monthly lowest groundwater level relative to the period 1989–2018.