# Peer review of "The 2018 northern European hydrological drought and its drivers in a historical perspective"

_Hydrology and Earth System Sciences, 2020_

## Referee Comment (RC1) · Anonymous Referee #1 · 9 Jul 2020

The manuscript "The 2018 northern European hydrological drought and its drivers in a historical perspective" by Sigrid J. Bakke et al. presents an analysis of the 2018 drought and its historical extremeness with a focus on northern European countries. With the analysis of the relationship between large-scale atmospheric circulations and summer streamflow, the assessment of the meteorological situation for the northern region and the resulting meteorological and hydrological drought on a region scale, the study gives a broad and detailed analysis of the 2018 drought for Northern Europe not only in different time- but also spatial scales. The study highlights that there is more need to assess drought in different components of the hydrological cycle, especially due to the complexity and large variability in drought characteristics that can be seen across different spatial scales and hydroclimatic regions, nicely show with this case

study for Norther Europe. In general I found the paper was well written and provided a good analysis of the 2018 drought. Therefore I would like to recommend publication after minor revisions. My comments for improvement can be found below.

Major comments:

The paper has a very clear structure and additional division of the assessment into different scales, making it clear which data and methods are used for which scale and analysis. The combination of datasets (including not only meteorological but also hydrological ones) on various scales gives the chance to assess the drought situation of 2018 for this region in more detail. The results of the analysis are explained and discussed in detail (which is good in general) but can lead to difficulties to follow all the information presented and taking away the key findings. Adding a small subchapter at the end of Section 5 with parts of the conclusion, where all the results are placed together, would help to connect the different discussion parts already earlier and leave more space for an even more concise conclusion. The figures used are nicely selected and interesting, especially Fig.8 including the groundwater response to precipitation and Fig.1 and 2 to highlight the streamflow and groundwater regimes, allowing the reader to get a better understanding of the hydroclimatological characteristics of the case area.

The introduction is giving an overview of the general drought situation and impacts for this region, elaborating on the study area and setting the stage for the study by recapping the general definition of drought, drought studies and their difficulties in regards to appropriate data selection and use. Further, a section on the large scale atmospheric drivers is giving, which is part of the later assessment. An additional elaboration on the other methods included and the reasoning behind using them would help prepare the reader for the following analysis and results and would strengthen the introduction and emphasizing why this paper is special in its own way and closing current research gaps. Adding more information on this and mentioning more similar studies might also help setting the scene for a deeper discussion later on.

The analysis is focused on the extremeness of the months May-August 2018, as mentioned in the abstract and introduction, highlighting the situation on conditions for northern European countries in that period. Despite stating the aim of the study clearly in the introduction, the title can lead to a slight misunderstanding. Nevertheless, having done such an extensive analysis of various aspects of the hydrological cycle for the whole year (as given by the information in the supplement), I personally think including some more lines on the results and observation in early spring until the end of the year, besides the extreme events observation in the period of May-August 2018, would create an even better base to start a wholesome discussion. Especially, as the findings are currently discussed within the light of the whole annual cycle (Sec.5.2) and it is mentioned that antecedent water storage (initial conditions) play an important role in the occurrence, timing and development of hydrological droughts and drought propagation. Extending the results and discussion to months where drought characteristics were also observed in April and autumn months (e.g. Fig A6 (SPI3), A7 (SPEI3), A9 and Fig.8 (groundwater ranks and groundwater response to precipitation)), could help to create an even better understanding of the drought situation of 2018. This in the end might help to create an even stronger discussion and to put the work into more context by being able to connect it to other drought studies of 2018 throughout Europe, bringing together other strains of research and closing the picture of the drought 2018.

Minor comments:

Table 1: adding an additional column for the observed impact category (e.g. agriculture, energy sector, etc.) would make table even more complete and could reduce effort to write all examples out in text;

p5 line21: 3 stations within mountain regimes mentioned which were highly influenced by glaciers, were they treated differently in the analysis or just included in the average?;

p5 line34: has instead of have (twice);

Data and methods section in general: focus on historical analysis: In regards to human

influence there was a careful selection of near natural groundwater wells but to what extend was climate change reconsidered in the analysis and the trend that might have been included automatically in the datasets used?;

Results and discussion section in general: also include beginning and end of the year results next to extremeness of summer months if mentioned later on in discussion (for example HGT500 from April might already indicate how situation in May could look like);

Fig. 4 and Fig. A3 using the same range for HGT500 values for all months presented would allow to compare values between months more easily. Additional question to Fig.4: why aggregate over May-August (as most other results presented are shown separately per month)?;

General comment on ranking system: nice to highlight extremes (as it is one of the goals mentioned in the introduction) but additional information and figures on mean historical temp vs 2018 temp would help to put this into place in regards to absolute values, also helps to understand precipitation observations as not that many low extremes were recognised but in SPI3 drought is indicated;

Fig10: what was the reasoning to switch to months June-August for this analysis, compared to the other results that have been heavily focused on period May-August?;

Discussion, section about annual hydrological cycle: more information and figures about initial conditions (e.g. snowfall) in supplement (e.g. annual averaged timeseries and 2018 situation, similar to Fig.1 and 2) and citations would support and help to follow the explanation of the specific observations and putting them into more context (some good starting information was already given in introduction about the hydroclimatological characteristics, streamflow and groundwater regimes);

p16 line2: citations or other examples to underline this assumption?;

p16 line14-16: could you elaborate a bit more (e.g. references to figures where this is

observed). If I look at Fig A9, A8, A7 for example I see overlapping areas and stations with indicate drought occurrence?;

p16 line24: would you say this is already the effect of drought propagation one can observe (with the ongoing dry conditions until the end of the year (e.g. seen in SPEI3 results)?;

p17 line8-9: maybe include this reference already in introduction to set the stage for the discussion; p 17 line25, spelling error: wells instead of well;

Appendix: A1 mountain regime: why not include December as winter month for classification criteria for streamflow regime?;

A1 line7: missing point after class

---

## Referee Comment (RC2) · Anonymous Referee #2 · 13 Aug 2020

Summary

This paper studies the May-August 2018 European drought from an atmospheric perspective, and meteorological and hydrological drought analyses. The authors found that record-breaking temperatures in May and July 2018 were observed in northern Europe regions accompanying meteorological drought denoted by SPEI-3 from May to July 2018. However, the hydrological drought shown by streamflow and groundwater drought started to develop in June and July, respectively. The author also found that local terrestrial processes including aquifer properties are important in controlling the hydrological drought response to meteorological conditions.

Assessment

[Figure]

This paper analyzes the 2018 northern European drought from different perspectives. The manuscript is interesting and well written. I have only a few minor comments below and most of them are for clarification. I believe this work is well suited for HESS.

Line by line comments

L refers to line and P refers to page.

P4L24: Do the temperature data here refer to 2 m temperature?

P4L31-32: I am wondering why do the authors use 2 different spatial scales for analyses in section 3.1 and 3.2 (0.25°), and 3.3 (0.1°)?. Why do not simply use a spatial resolution of 0.1°?

P8L15: The authors may write: three-month.

P8L27-29: Here, I am also wondering why do the authors use SPI-3 (SPEI-3) distributions derived from the data year 1971 to 2000 to calculate SPI-3 (SPEI-3) in the year 2018? Why do not use the distribution derived from 1971 to present data? By only using data from 1971 to 2000 (20 years ago), the drought 2018 might be too extreme because the authors excluded extreme drought years e.g. 2003, 2006-2008, and 2015. This has implications in the distributions that the authors used. Moreover, the average temperature >20 years ago was lower than the average temperature in the past 20 years (2000-2020). In Europe, we also use drought years 1976 and 2003 as a benchmark for extreme drought years. 2018 was comparable to those years in terms of drought severity. This question applies to other reference data (e.g. section 3.1, from 1981 to 2000).

P9L4-6: I am wondering why do the authors use absolute values to determine the SPI classes? Figure 6 also shows the SPI/SPEI index values from -3 to +3.

P10L3: The authors may write as Figure 3a-d.

P10L11: The authors may write as Figure 3e-h.

[Figure]

P12L30: Please write the Figure number after the sentence thus the reader can follow the description easily. Here is Figure 9a.

P12L33: The authors may write Figure 9b after the sentence.

P13L2: The authors may write Figure 9c after the sentence.

P14L20: Typo "than 3 std, respectively 2 std"

P24: Table 1: The author may write last accessed before the date. E.g. (last accessed 24.03.20).

P25: Back to my question about the reference data, here in Table 2, the authors indicate that they have temperature, precipitation, Geopotential height at 500MB data up to the year 2018.
* * *

---

## Author Comment (AC1) · 24 Aug 2020

Thank you very much for the positive and constructive feedback on our paper "The 2018 northern European hydrological drought and its drivers in a historical perspective". Hereby, we would like to respond to your comments:

All reviewer's comments regarding language corrections will be accounted for in the revision. Other more substantial comments are responded to below, with the original comments marked by 'AR#1' and our response paragraphs marked by 'Authors'.

AR#1: Adding a small subchapter at the end of Section 5 with parts of the conclusion, where all the results are placed together, would help to connect the different discussion parts already earlier and leave more space for an even more concise conclusion.

[Figure]

Authors: The structure of the discussion chapter was something we discussed extensively during the writing process, in particular the discussion related to drought propagation. Currently the discussion regarding 2018 drought propagation is embedded in Sect. 5.2. Following your suggestion, we will consider adding a new subsection at the end of Section 5 bringing together the key results in the context of drought propagation by moving parts of the content from the conclusion and Sect. 5.2.

AR#1: In the introduction: An additional elaboration on the other methods included and the reasoning behind using them would help prepare the reader for the following analysis and results and would strengthen the introduction and emphasizing why this paper is special in its own way and closing current research gaps. Adding more information on this and mentioning more similar studies might also help setting the scene for a deeper discussion later on.

Authors: We agree on including a more complete presentation of the methods applied, including their motivation as well as potential similar studies not already mentioned in the introduction. We will embed this in our revised version.

AR#1: I personally think including some more lines on the results and observation in early spring until the end of the year, besides the extreme events observation in the period of May-August 2018, would create an even better base to start a wholesome discussion. (. . .) Extending the results and discussion to months where drought characteristics were also observed in April and autumn months. (. . .) This in the end might help to create an even stronger discussion and to put the work into more context by being able to connect it to other drought studies of 2018 throughout Europe, bringing together other strains of research and closing the picture of the drought 2018.

Authors: We agree that it is a good idea to include some more lines about the spring and autumn to make a more wholesome discussion, and we will do that in our revised version. If you are aware of other studies of the 2018 drought in Europe, we are happy to receive a note on that. Regardless of already existing studies, we find your

comment to extend the discussion beyond May-August 2018, valuable in that potential future studies on the 2018 drought can more easily connect it to our paper if accepted.

AR#1: Table 1: adding an additional column for the observed impact category (e.g. agriculture, energy sector, etc.) would make table even more complete and could reduce effort to write all examples out in text.

Authors: We agree it is a good idea to include the impact category in Table 1, and we will do so in our revised version following the EDII categorisation provided in Stahl et al. (2016; doi:10.5194/nhess-16-801-2016). We will further ensure that the format of the Table fits nicely to one page, despite its extension, and potentially reduce details on the impacts in the text.

AR#1: p5 line21: 3 stations within mountain regimes mentioned which were highly influenced by glaciers, were they treated differently in the analysis or just included in the average?

Authors: We are not sure which average you refer to here, but the regimes highly influenced by glaciers were not treated differently from the other regimes in the analysis. Accordingly, the stations are included in the total percentages of stations affected and in the EOF analysis in the same way as the other stations (also reflecting widely different regimes).

AR#1: (...) to what extend was climate change reconsidered in the analysis and the trend that might have been included automatically in the datasets used?

Authors: Climate change was not considered explicitly in the analysis of the 2018 drought, and accordingly, potential trends in both the average and extreme conditions are automatically included. The main purpose of the ranking maps was to investigate the extremeness of individual months in 2018 compared to previously observed. The paper do not address the cause of the extreme ranking itself, rather whether or not it was an extreme year as compared to the historical record. On the other hand, the

purpose of the EOF analysis was to detect main patterns in summer streamflow variability, and linear detrending of the JJA streamflow time series was conducted prior to the EOF calculation (ref. p9, line 27-28).

AR#1: Results and discussion section in general: also include beginning and end of the year results next to extremeness of summer months if mentioned later on in discussion (for example HGT500 from April might already indicate how situation in May could look like);

Authors: We interpret this comment as being related to the third comment, both which we agree to.

AR#1: Fig. 4 and Fig. A3 using the same range for HGT500 values for all months presented would allow to compare values between months more easily. Additional question toFig.4: why aggregate over May-August (as most other results presented are shown separately per month)?

Authors: We did not seek to have the same scale on the HGT500 axes; rather focus was on depicting the relative variability for each month using standard deviation (thus allowing direct comparison of the variability as such). Figure 1 below shows the same figure as Fig. A3 in the paper, except that the same range is used for HGT500 values in the right panel. Accordingly, the time series shifts its location (up or down) along the HGT500 axis. We do not see any advantage of presenting the results in this way, and prefer to keep the figure as it is. Nevertheless, we will add a remark that the range is different, to ease the interpretation for the reader. Figure 4 is provided to emphasis the extreme overall large-scale atmospheric situation in the period May-August. Combined with separate monthly plots in the Appendix (Fig. A3), we think this provide an informative overview for the reader.

AR#1: (...) additional information and figures on mean historical temp vs 2018 temp would help to put this into place in regards to absolute values, also helps to understand precipitation observations as not that many low extremes were recognised but in SPI3

drought is indicated;

Authors: We agree that this is interesting additional information. We will make anomaly maps of temperature and precipitation and add these to the appendix or supplement.

AR#1: Fig10: what was the reasoning to switch to months June-August for this analysis, compared to the other results that have been heavily focused on period May-August?

Authors: The main reason to use June-August instead of May-August in the composite maps, was to use the same period of the year as used for the EOF analysis of streamflow that the composites are based on. The main reason for using June-August instead of May-August in the EOF analysis was to avoid the effect of high flow in May caused by snowmelt. Furthermore, EOF analysis and composite maps are traditionally done on a seasonal basis, making the results more easily comparable to other studies. We will make the reasoning behind the choice of June-August more clear in the text in our revised version.

AR#1: Discussion, section about annual hydrological cycle: more information and figures about initial conditions (e.g. snowfall) in supplement (e.g. annual averaged time series and 2018 situation, similar to Fig.1 and 2) and citations would support and help to follow the explanation of the specific observations and putting them into more context (some good starting information was already given in introduction about the hydroclimatological characteristics, streamflow and groundwater regimes);

Authors: Observed annual average time series plotted along with the 2018 time series for each streamflow and groundwater station, were made as part of the initial analysis, but not included in the paper itself due to the article already being relatively long. We agree that they can help support the interpretation and discussion and suggest to include such figures in the appendix or supplement. We will not include data of initial conditions, such as snow and soil moisture for each catchment, as this would require using modelled data.

AR#1: p16 line2: citations or other examples to underline this assumption?

Authors: This is a speculation based on the known important role of groundwater in Denmark and the less extreme potential ET losses in the west compared to the east (mentioned in the previous sentence, ref. Fig. 6 in the paper). We will include references to underline this assumption in the revised version.

AR#1: p16 line14-16: could you elaborate a bit more (e.g. references to figures where this is observed). If I look at Fig A9, A8, A7 for example I see overlapping areas and stations with indicate drought occurrence?

Authors: We will refer more specifically to figures used in this assessment, and clarify what we mean in the text.

AR#1: p16 line24: would you say this is already the effect of drought propagation one can observe (with the ongoing dry conditions until the end of the year (e.g. seen in SPEI3 results)?

Authors: We interpret your question as to whether the below normal groundwater levels at end 2018 /start 2019 are a response to the summer 2018 event or caused by the dry conditions towards the end of the year. It is probably a combined effect of the two. Following the inclusion of the situation in the spring and autumn (ref. third comment), we will embed an assessment related to this comment in the revised version.

AR#1: p17 line8-9: maybe include this reference already in introduction to set the stage for the discussion;

Authors: We will try to find a natural place for this reference in the introduction.

AR#1: Appendix: A1 mountain regime: why not include December as winter month for classification criteria for streamflow regime?

Authors: The reason for this is that none of the stations have minimum or a second minimum flow in December (the same is true for November). We will add a note about

this in Appendix A1. November or December is typically the beginning of the winter season, whereas lowest flow for stations with winter low flow regime typically occurs towards the end of the winter season (most winter low flow regime stations have minimum flow in February and March).

[Figure]

**HGT500 subdomain standard deviation**

**Fig. 1.** Same as Figure A3 in the paper, except that the same range is used for HGT500 values in the right panel.

---

## Author Comment (AC2) · 2 Sep 2020

Thank you so much for the helpful feedback on our paper "The 2018 northern European hydrological drought and its drivers in a historical perspective". Hereby, we would like to respond to your comments (comments marked by 'AR#2' and response paragraphs marked by 'Authors'):

AR#2: P4L24: Do the temperature data here refer to 2 m temperature?

Authors: The E-OBS temperature data is interpolated station data of air temperature, which to our knowledge are measurements at 2 m (currently getting this confirmed).

AR#2: P4L31-32: I am wondering why do the authors use 2 different spatial scales for analyses in section 3.1 and 3.2 (0.25), and 3.3 (0.1)?. Why do not simply use a spatial

resolution of 0.1?

Authors: We agree that this is confusing, and will change to use the resolution of 0.1 for all analysis using the E-OBS datasets. The figures will be remade using the resolution of 0.1 degrees.

AR#2: P8L15: The authors may write: three-month.

Authors: We will do so in the revised version.

AR#2: P8L27-29: Here, I am also wondering why do the authors use SPI-3 (SPEI-3) distributions derived from the data year 1971 to 2000 to calculate SPI-3 (SPEI-3) in the year 2018? Why do not use the distribution derived from 1971 to present data? By only using data from 1971 to 2000 (20 years ago), the drought 2018 might be too extreme because the authors excluded extreme drought years e.g. 2003, 2006-2008, and 2015. This has implications in the distributions that the authors used. Moreover, the average temperature >20 years ago was lower than the average temperature in the past 20 years (2000-2020). In Europe, we also use drought years 1976 and 2003 as a benchmark for extreme drought years. 2018 was comparable to those years in terms of drought severity. This question applies to other reference data (e.g. section 3.1, from 1981 to 2000).

Authors: We agree that the extremeness of the anomaly plots as well as SPI3 and SPEI3 are sensitive to the choice of reference period. The reason we use a 30-year period of reference and not the period 1971 to 2018 is to allow for easier comparison with other studies (e.g. lonita et al., 2017; doi:10.5194/hess-21-1397-2017). Even though the 30-year period of reference might be subject to choice, they are more consistent than using a longer period up to the year of interest. However, we used the ranking maps to be able to investigate the historical extremeness compared to other extreme years during the whole 60-year period. The main purpose of including the SPI and SPEI figures is to map the dynamic (in space and time) of the meteorological drought. Following the reviewer's remark, we calculated the SPI and SPEI using
the whole period (1959-2018) as reference, and found similar spatial patterns in the drought evolution throughout 2018 (ref. monthly plots). See attached figures at the end of the document. Accordingly, we prefer to keep the 30-year period of reference (i.e. 1971-2000).

AR#2: P9L4-6: I am wondering why do the authors use absolute values to determine the SPI classes? Figure 6 also shows the SPI/SPEI index values from -3 to +3.

Authors: If we understand your comment correctly, the confusion may arise from displaying absolute values rather than the wet and dry ranges. We separate between negative and positive values in the study, and suggest instead to write e.g. "defining SPI values in the range |1-1.5| (9.2 % probability) as moderate drought/moderately wet".

AR#2: P10L3: The authors may write as Figure 3a-d. P10L11: The authors may write as Figure 3e-h. P12L30: Please write the Figure number after the sentence thus the reader can follow the description easily. Here is Figure 9a. P12L33: The authors may write Figure 9b after the sentence. P13L2: The authors may write Figure 9c after the sentence.

Authors: We agree on the above five comments about figure number inclusions, and will do so in our revised version.

AR#2: P14L20: Typo "than 3 std, respectively 2 std" Authors: We are not sure what typo you refer to. It might be the use of commas. We will check this phrase with a native speaker.

AR#2: P24: Table 1: The author may write last accessed before the date. E.g. (last accessed 24.03.20).

Authors: "URL (last access)" is written in the column heading to indicate that the date in parenthesis is the last access date.

AR#2: P25: Back to my question about the reference data, here in Table 2, the authors
indicate that they have temperature, precipitation, Geopotential height at 500MB data up to the year 2018.

Authors: Yes, Table 2 shows the data used for the different indices. If we are to change the reference period of parts of the analysis, we will also update the Table accordingly.

---

## Author Response (AR1)

Dear Editor,

Thank you for taking your time to handle and report on our manuscript. Hereby, we would like to provide our point-by-point reply to the comments of Anonymous Referee #1 (AR#1) and Anonymous Referee #2 (AR#2). Finally, we also respond to the remark from the editor.

Original comments are marked by the referee abbreviation 'AR#1' or 'AR#2', the remark by the editor by 'Editor', our responses by 'Authors' and reference to the places where changes have been made in the track-changes version of the manuscript are marked by 'Change'. In addition to the changes following the comments by AR#1, AR#2 and Editor, we have made minor text editions to correct spelling/grammar or increase readability. All text changes are visible in the track-changes version of the manuscript. Note that the track-changes file does not mark changes for remade figures. The figures that have been remade are Fig. 3, 5, 6, 8, A1, A4, A5, A6 and A7.

**Response to comments by Anonymous Referee #1**

| | | |
|---|---|---|
| 1.1 | AR#1 | The paper has a very clear structure and additional division of the assessment into different scales, making it clear which data and methods are used for which scale and analysis. The combination of datasets (including not only meteorological but also hydrological ones) on various scales gives the chance to assess the drought situation of 2018 for this region in more detail. The results of the analysis are explained and discussed in detail (which is good in general) but can lead to difficulties to follow all the information presented and taking away the key findings. Adding a small subchapter at the end of Section 5 with parts of the conclusion, where all the results are placed together, would help to connect the different discussion parts already earlier and leave more space for an even more concise conclusion. The figures used are nicely selected and interesting, especially Fig.8 including the groundwater response to precipitation and Fig.1 and 2 to highlight the streamflow and groundwater regimes, allowing the reader to get a better understanding of the hydroclimatological characteristics of the case area. |
| | Authors | The structure of the discussion chapter was something we discussed extensively during the writing process, in particular the discussion related to drought propagation. Currently the discussion regarding 2018 drought propagation is embedded in Sect. 5.2. Following the suggestion, we considered adding a new subsection at the end of Section 5 bringing together the key results in the context of drought propagation by moving parts of the content from the conclusion and Sect. 5.2. However, we find the original structure of the discussion the better option, in which the drought propagation becomes a natural part of the discussion of the hydrological aspect of the drought. |
| | Change | No change in manuscript. |
| 1.2 | AR#1 | The introduction is giving an overview of the general drought situation and impacts for this region, elaborating on the study area and setting the stage for the study by recapping the general definition of drought, drought studies and their difficulties in regards to appropriate data selection and use. Further, a section on the large scale atmospheric drivers is giving, which is part of the later assessment. An additional elaboration on the other methods included and the reasoning behind using them would help prepare the reader for the following analysis and results and would strengthen the introduction and emphasising why this paper is special in its own way and closing current |

|  |  | research gaps. Adding more information on this and mentioning more similar studies might also help setting the scene for a deeper discussion later on. |
|---|---|---|
|  | Authors | We agree on including a more complete presentation of the methods applied, including their motivation as well as potential similar studies not already mentioned in the introduction. We have embedded this in our revised version. |
|  | Change | Page 4: line 21-34. |

| 1.3 | AR#1 | The analysis is focused on the extremeness of the months May-August 2018, as mentioned in the abstract and introduction, highlighting the situation on conditions for northern European countries in that period. Despite stating the aim of the study clearly in the introduction, the title can lead to a slight misunderstanding. Nevertheless, having done such an extensive analysis of various aspects of the hydrological cycle for the whole year (as given by the information in the supplement), I personally think including some more lines on the results and observation in early spring until the end of the year, besides the extreme events observation in the period of May-August 2018, would create an even better base to start a wholesome discussion. Especially, as the findings are currently discussed within the light of the whole annual cycle (Sec.5.2) and it is mentioned that antecedent water storage (initial conditions) play an important role in the occurrence, timing and development of hydrological droughts and drought propagation. Extending the results and discussion to months where drought characteristics were also observed in April and autumn months (e.g. Fig A6 (SPI3), A7 (SPEI3), A9 and Fig.8 (groundwater ranks and groundwater response to precipitation)), could help to create an even better understanding of the drought situation of 2018. This in the end might help to create an even stronger discussion and to put the work into more context by being able to connect it to other drought studies of 2018 throughout Europe, bringing together other strains of research and closing the picture of the drought 2018. |
|---|---|---|
|  | Authors | We agree that it is a good idea to include some more lines about the early spring until the end of the year, and we have done this in our revised version. We find your comment to extend the discussion beyond May-August 2018, valuable in that potential future studies on the 2018 drought can more easily connect it to our paper if accepted. |
|  | Change | Page 11: line 10-11, 22-23, 29-31. Page 12: line 25-27, 34-35. Page 13: line 1-2, 5-7, 12-19, 30-31. Page 16: line 27. Page 17: line 30-31. Page 18: line 19-22, 35. Page 19: line 1-2. |

| 1.4 | AR#1 | Table 1: adding an additional column for the observed impact category (e.g. agriculture, energy sector, etc.) would make table even more complete and could reduce effort to write all examples out in text |
|---|---|---|
|  | Authors | We agree it is a good idea to include the impact category in Table 1, and we have done so in our revised version following the EDII categorisation provided in Stahl et al. (2016; doi:10.5194/nhess-16-801-2016). We also changed parts of the set-up of the table (mainly the order of the columns) to make it more easily readable. In the discussion format of the paper, the table is larger than one page, however, in the two-column format, it fits nicely into one page (see snap shot at the end of this document). If the former is a problem, we can optionally move the URLs to another location, e.g. by making a list of URLs after the reference list. We could not find any house rules related to handling URLs, so if we are to change the format of the table and potentially move the URLs, specific guidelines on how this is preferably done, are appreciated. |
|  | Change | Table 1 on page 27. |

| 1.5 | AR#1 | p5 line21: 3 stations within mountain regimes mentioned which were highly influenced by glaciers, were they treated differently in the analysis or just included in the average? |
| | Authors | We are not sure which average is referred to here, but the regimes highly influenced by glaciers were not treated differently from the other regimes in the analysis. Accordingly, the stations are included in the total percentages of stations affected and in the EOF analysis in the same way as the other stations (also reflecting widely different regimes). |
| | Change | No change in manuscript. |

| 1.6 | AR#1 | p5 line34: has instead of have (twice) |
| | Authors | This has been corrected. |
| | Change | Page 6: line 25. |

| 1.7 | AR#1 | Data and methods section in general: focus on historical analysis: In regards to human influence there was a careful selection of near natural groundwater wells but to what extend was climate change reconsidered in the analysis and the trend that might have been included automatically in the datasets used? |
| | Authors | Climate change was not considered explicitly in the analysis of the 2018 drought, and accordingly, potential trends in both the average and extreme conditions are automatically included. The main purpose of the ranking maps was to investigate the extremeness of individual months in 2018 as compared to the historical record. On the other hand, the purpose of the EOF analysis was to detect main patterns in summer streamflow variability, and linear detrending of the JJA streamflow time series was conducted prior to the EOF calculation (ref. p9, line 27-28). |
| | Change | No change in manuscript. |

| 1.8 | AR#1 | Results and discussion section in general: also include beginning and end of the year results next to extremeness of summer months if mentioned later on in discussion (for example HGT500 from April might already indicate how situation in May could look like) |
| | Authors | We agree, and interpret this comment as being related to comment 1.3 and 1.15. |
| | Change | See reply to 1.3 and 1.15 |

| 1.9 | AR#1 | Fig. 4 and Fig. A3 using the same range for HGT500 values for all months presented would allow to compare values between months more easily. Additional question to Fig.4: why aggregate over May-August (as most other results presented are shown separately per month)? |
| | Authors | We did not seek to have the same scale on the HGT500 axes; rather focus on depicting the relative variability for each month using standard deviation (thus allowing direct comparison of the variability as such). We made a figure for our online response to AR#1 showing the same figure as Fig. A3 in the paper, except that the same range is used for HGT500 values in the right panel. Accordingly, the time series shifts its location (up or down) along the HGT500 axis. We do not see any advantage of presenting the results in this way, and prefer to keep the figure as it is. Nevertheless, we will add a remark that the range is different, to ease the interpretation for the reader. Figure 4 is provided to emphasise the extreme overall large-scale atmospheric situation in the |

period May-August. Combined with separate monthly plots in the Appendix (Fig. A3), we think this provides an informative overview for the reader.

| | | |
|---|---|---|
| | Change | Page 41, added sentence in the Fig. A3 caption: *"Note the different ranges of the y-axes."* |

| | | |
|---|---|---|
| 1.10 | AR#1 | General comment on ranking system: nice to highlight extremes (as it is one of the goals mentioned in the introduction) but additional information and figures on mean historical temp vs 2018 temp would help to put this into place in regards to absolute values, also helps to understand precipitation observations as not that many low extremes were recognised but in SPI3 drought is indicated |
| | Authors | We agree that this is interesting additional information. We have made anomaly maps for each month in 2018 of temperature and precipitation and added these to the supplement. |
| | Change | Supplement Figure S1 and S2. Referred to in the main text on:  Page 12: line 10, 29. Page 18: line 12, 20. |

| | | |
|---|---|---|
| 1.11 | AR#1 | Fig10: what was the reasoning to switch to months June-August for this analysis, compared to the other results that have been heavily focused on period May-August? |
| | Authors | The main reason to use June-August instead of May-August in the composite maps (Fig. 10), was to use the same period as used for the EOF analysis of streamflow that the composites are based on. The main reason for using June-August instead of May-August in the EOF analysis was to avoid the effect of high flow in May caused by snowmelt. Furthermore, EOF analysis and composite maps are traditionally done on a three-month seasonal basis, making the results more easily comparable to other studies. We have made the reasoning behind the choice of June-August more clear in the text in our revised version. |
| | Change | Page 10: line 31-33. Page 11: line 1. We have added two sentences: *"The June-August period was chosen for the EOF analysis (rather than May-August, which is in focus in Sect. 3.1-3.3) to avoid the effect of high flow in May caused by snowmelt. Furthermore, EOF analysis and composite maps are traditionally done on a three-month seasonal basis, making the results more easily comparable to other studies."* |

| | | |
|---|---|---|
| 1.12 | AR#1 | Discussion, section about annual hydrological cycle: more information and figures about initial conditions (e.g. snowfall) in supplement (e.g. annual averaged timeseries and 2018 situation, similar to Fig.1 and 2) and citations would support and help to follow the explanation of the specific observations and putting them into more context (some good starting information was already given in introduction about the hydroclimatological characteristics, streamflow and groundwater regimes) |
| | Authors | Observed annual average time series plotted along with the 2018 time series for each streamflow and groundwater station, were made as part of the initial analysis, but not included in the paper itself due to the article already being relatively long. We agree that they can help support the interpretation and discussion. We have now made figures of standardised monthly  average streamflow and groundwater levels in 2018 vs multiyear monthly statistics for each station separately, and added them to the supplement. We will not include data of initial conditions, such as snow and soil moisture for each catchment, as this would require using modelled data. |

|         | Change  | Supplement Figure S3—S5 (streamflow) and Figure S6—7 (groundwater). Referred to in the main text on: Page 6: line 11. Page 7: line 3. Page 13: line 27, 33. Page 14: line 1. Page 17: line 23, 28. Page 18, line 15, 31. |

| 1.13 | AR#1 | p16 line2: citations or other examples to underline this assumption? |
| | Authors | We removed the part of the sentence where we speculate about the role of persistent groundwater contribution, to connect the sentence directly to the argument of the previous sentence. |
| | Change | The text now reads (page 18: line 4-8): *"A southeastern-northwestern gradient in extreme temperature (and SPEI3) this month, however, reflects the spatial pattern of extremely low streamflow in Denmark, indicating that higher than usual evapotranspiration rates likely contributed to extreme conditions in the southeast. Correspondingly, less extreme evapotranspiration in the west and north might have prevented streamflow drought to develop there."* |

| 1.14 | AR#1 | p16 line14-16: could you elaborate a bit more (e.g. references to figures where this is observed). If I look at Fig A9, A8, A7 for example I see overlapping areas and stations with indicate drought occurrence? |
| | Authors | We agree that there are overlapping areas and stations which indicate drought occurrence. Our point about the high local variability was that several wells have no rank 1-6 at locations where other wells have rank 1-6. We have clarified what we mean in the text. |
| | Change | The sentence now reads (page 18: line 26-28): *"The high spatial variability in hydrogeological properties across the Nordic region is mirrored in the diversity in groundwater response to meteorological conditions, as reflected in a high local variability for groundwater drought (rank between 1 and 6) even for closely located wells."* |

| 1.15 | AR#1 | p16 line24: would you say this is already the effect of drought propagation one can observe (with the ongoing dry conditions until the end of the year (e.g. seen in SPEI3 results)? |
| | Authors | We interpret your question as to whether the below normal groundwater levels at end 2018 /start 2019 are a response to the summer 2018 event or caused by dry conditions following the below normal rainfall September-November 2018. It is probably a combined effect of the two. Following the inclusion of the situation in the spring and autumn (ref. third comment), we will embed an assessment of both the streamflow and groundwater conditions related in the revised version. |
| | Change | Page 18: line 19-22, 35. Page 19: 1-2. |

| 1.16 | AR#1 | p17 line8-9: maybe include this reference already in introduction to set the stage for the discussion |
| | Authors | We were unable to find a natural place for these references in the introduction without adding a new paragraph related to assessments of historical drought events in Europe. We prefer to not increase the size of the (in our opinion) already long introduction in order to include these references there. |
| | Change | No change in manuscript. |

| 1.17 | AR#1 | p 17 line25, spelling error: wells instead of well. |
| | Author | This has been corrected. |
| | Change | Page 20: line 9. |

| 1.18 | AR#1 | Appendix: A1 mountain regime: why not include December as winter month for classification criteria for streamflow regime? |
| | Authors | The reason is that none of the stations have minimum or a second minimum flow in December (the same is true for November). We will add a note about this in Appendix A1. November/December is typically the beginning of the winter season, and lowest flow for stations with winter low flow regime typically occurs towards the end of the winter season (most winter low flow regime stations have minimum flow in February and March). |
| | Change | Page 37: line 13-14. Added the sentence: *"November and December were not included in the low flow season because none of the streamflow stations had the minimum or second minimum flow in these months."* |

| 1.19 | AR#1 | A1 line7: missing point after class |
| | Authors | This has been corrected. |
| | Change | Page 37: line 7. |

**Response to comments by Anonymous Referee #2**

| 2.1 | AR#2 | P4L24: Do the temperature data here refer to 2 m temperature? |
| | Authors | The E-OBS temperature data is interpolated station data of air temperature. We asked this question to the E-OBS project team. They answered that temperature is not always measured at 2 meters by all data providers, they do not know the exact measuring height for all data providers, and they do not correct measurements to have a 'standard' height. |
| | Change | No change in manuscript. |

| 2.2 | AR#2 | P4L31-32: I am wondering why do the authors use 2 different spatial scales for analyses in section 3.1 and 3.2 (0.25°), and 3.3 (0.1°)? Why do not simply use a spatial resolution of 0.1°? |
| | Authors | We agree that this is confusing, and in the revised version of the manuscript we use the resolution of 0.1 for all analysis and figures using the E-OBS dataset. Since the E-OBS dataset has been updated after we made the figures used in the original manuscript, we have used the newest available version (v21.0e) when remaking the figures. The percentages of grid cells with SPI3 and SPEI3 <-1.5 are updated according to the new dataset. |
| | Change | Figure 5 (page 31), 6 (page 32), 8 (page 34), A4 (page 42), A5 (page 43), A6 (page 44) and A7 (page 45) are remade, and Figure S1 and S2 in the supplement are made, using the E-OBS dataset v21.0e at 0.1deg resolution. Data description changed on page 5: line 16-18. Percentages changed on page 13: line 8, 21-22. |

| 2.3 | AR#2 | P8L15: The authors may write: three-month. |
| | Authors | This has been corrected. |
| | Change | Page 9: line 13, 25. Page 28: two times in Table 2. |

| 2.4 | AR#2 | P8L27-29: Here, I am also wondering why do the authors use SPI-3 (SPEI-3) distributions derived from the data year 1971 to 2000 to calculate SPI-3 (SPEI-3) in the year 2018? Why do not use the distribution derived from 1971 to present data? By only using data from 1971 to 2000 (20 years ago), the drought 2018 might be too extreme because the authors excluded extreme drought years e.g. 2003, 2006-2008, and 2015. This has implications in the distributions that the authors used. Moreover, the average temperature >20 years ago was lower |

|  |  |  |
|---|---|---|
|  |  | than the average temperature in the past 20 years (2000-2020). In Europe, we also use drought years 1976 and 2003 as a benchmark for extreme drought years. 2018 was comparable to those years in terms of drought severity. This question applies to other reference data (e.g. section 3.1, from 1981 to 2000). |
|  | Authors | We agree that the extremeness of the anomaly plots as well as SPI3 and SPEI3 can be sensitive to the choice of reference period. The reason we use a 30-year period of reference (ref. WMO guidelines) and not the period 1971 to 2018 is to allow for easier comparison with other studies (e.g. Ionita et al., 2017; doi:10.5194/hess-21-1397-2017). Even though a 30-year period of reference might be subject to choice, it is more consistent than using a longer period up to the year of interest. A key focus of our study was to use the ranking maps to investigate the historical extremeness compared to other extreme years during the whole 60-year period. The main purpose of including the SPI and SPEI figures was to map the dynamic (in space and time) of the meteorological drought. Following the reviewer's remark, we calculated the SPI and SPEI using the whole period (1959-2018) as reference (figures in our online response to AR#2), and found similar spatial patterns in the drought evolution throughout 2018 (ref. monthly plots). Accordingly, we prefer to keep the 30-year period of reference (i.e. 1971-2000).

The SST reference period (originally 1981-2000) was the closest we could get 1971 to 2000 due to shortage of data. However, we agree that it is beneficial to use the same period of reference for all analysis. We have chosen to use the monthly SST data from the Hadley Centre (HadISST) for the SST anomaly calculations. The HadISST dataset has a coarser spatial resolution, but we were able to use the reference period 1971-2000. We have remade the SST figures using the HadISST data and 1971-2000 as reference period. |
|  | Change | Figure 3 (page 29) and Figure A1 (page 39) are remade using the HadISST dataset and the ref. period 1971-2000. Reference period changed on page 7: line 2, page 11: line 9, page 28 (in Table 2), page 29 (in figure caption), page 39 (in figure caption). Results changed on page 1: line 7, page 11: line 16. Data description changed on page 5: line 12-14, page 22: line 1-2. |
| 2.5 | AR#2 | P9L4-6: I am wondering why do the authors use absolute values to determine the SPI classes? Figure 6 also shows the SPI/SPEI index values from -3 to +3. |
|  | Authors | If we understand your comment correctly, the confusion may arise from displaying absolute values rather than the wet and dry ranges. We separate between negative and positive values in the study, and have clarified this in the revised version. |
|  | Change | Page 10: line 1-6. |
| 2.6 | AR#2 | P10L3: The authors may write as Figure 3a-d. |
|  | Authors | We have included this in the revised version. |
|  | Change | Page 11: line 9 |
| 2.7 | AR#2 | P10L11: The authors may write as Figure 3e-h. |
|  | Authors | We have included this in the revised version. |
|  | Change | Page 11: line 19. |
| 2.8 | AR#2 | P12L30: Please write the Figure number after the sentence thus the reader can follow the description easily. Here is Figure 9a. |

|       | Authors | We have included the figure numbers in the revised version. We also added reference to Figure 9d, 9e and 9f where appropriate. |
|       | Change  | Page 14: line 24, 26, 29, 32. |

| 2.9  | AR#2    | P12L33: The authors may write Figure 9b after the sentence. |
|      | Authors | We have included this in the revised version. |
|      | Change  | Page 14: line 27. |

| 2.10 | AR#2    | P13L2: The authors may write Figure 9c after the sentence. |
|      | Authors | We have included this in the revised version. |
|      | Change  | Page 14: line 30. |

| 2.11 | AR#2    | P14L20: Typo "than 3 std, respectively 2 std" |
|      | Authors | We are not sure what typo it is referred to. We see that 'std' jumped to the next line, and have now forced it to be on the same line as the values. |
|      | Change  | Page 16: line 17. |

| 2.12 | AR#2    | P24: Table 1: The author may write last accessed before the date. E.g. (last accessed 24.03.20). |
|      | Authors | "URL (last access)" is written in the column heading to indicate that the date in parenthesis is the last access date. To make the table more clear, and adding impact category after comment 1.4, the structure of the table has been changed. We have also changed from "last access" to "last accessed". |
|      | Change  | Table 1, page 27. |

| 2.13 | AR#2    | P25: Back to my question about the reference data, here in Table 2, the authors indicate that they have temperature, precipitation, Geopotential height at 500MB data up to the year 2018. |
|      | Authors | Yes, Table 2 shows the data used for the different indices. Currently the reference period 1971-2000 is kept (ref. answer to comment 2.4). We have updated the table following the change in reference period for the SST anomaly calculation (ref. answer to comment 2.4). |
|      | Change  | Period of reference for SST anomalies changed in Table 2 (page 28). |

**Editor's remark**

| 3.1 | Editor  | Concerning the remark of one of the authors on other studies on the 2018 drought the following: our group also has a manuscript in discussion (https://hess.copernicus.org/preprints/hess-2020-358/, feel free to comment), and there will be a special issue in Phil. Trans. R. Soc. B, see: https://royalsocietypublishing.org/rstb/forthcoming-issues. If it comes out soon it might be useful to you in the revision. |
|     | Authors | We thank the editor for good suggestions of 2018 papers, related to comment 1.3. The first suggestion concerns an interesting study of the 2018 drought's effects on root water uptake and a quantification of the critical moisture content in the Netherlands. We included this reference in the paragraph concerning energy-limited vs water-limited regions. The other suggestion is the now available special issue about the 2018 drought/heatwave impacts (https://royalsocietypublishing.org/toc/rstb/375/1810). Our impression is that the main focus is on the impact on terrestrial ecosystems (including crops). As |

we already have examples of those impacts in the introduction, we choose to not include them in the revised version.

Change        Page 17: line 3.

[revised manuscript text omitted]